# Unusual scaling laws for plasmonic nanolasers beyond the diffraction limit

Suo Wang[1], Xing-Yuan Wang[1], Bo Li[1], Hua-Zhou Chen[1], Yi-Lun Wang[1], Lun Dai [1,2], Rupert F. Oulton [3] & Ren-Min Ma[1,2]

Plasmonic nanolasers are a new class of amplifiers that generate coherent light well below the diffraction barrier bringing fundamentally new capabilities to biochemical sensing, super-resolution imaging, and on-chip optical communication. However, a debate about whether metals can enhance the performance of lasers has persisted due to the unavoidable fact that metallic absorption intrinsically scales with field confinement. Here, we report plasmonic nanolasers with extremely low thresholds on the order of 10 kW cm$^{-2}$ at room temperature, which are comparable to those found in modern laser diodes. More importantly, we find unusual scaling laws allowing plasmonic lasers to be more compact and faster with lower threshold and power consumption than photonic lasers when the cavity size approaches or surpasses the diffraction limit. This clarifies the long-standing debate over the viability of metal confinement and feedback strategies in laser technology and identifies situations where plasmonic lasers can have clear practical advantage.

[1] State Key Lab for Mesoscopic Physics and School of Physics, Peking University, Beijing 100871, China. [2] Collaborative Innovation Center of Quantum Matter, Beijing 100871, China. [3] The Blackett Laboratory, Department of Physics, Imperial College London, Prince Consort Road, London SW7 2AZ, UK. Suo Wang and Xing-Yuan Wang contributed equally to this work. Correspondence and requests for materials should be addressed to R.-M.M. (email: renminma@pku.edu.cn)

In the past four decades, the success of laser miniaturization is exemplified by the rapid development of vertical-cavity surface emitting lasers, microdisk lasers, photonic crystal lasers and nanowire lasers, with the goals of faster coherent light sources with lower power consumption[1–4]. These semiconductor lasers employ ever smaller microscale cavities to enhance the light matter interaction by which the laser modulation speed becomes faster and the threshold and power consumption become lower. While pure dielectric micro-cavities can reach scales approaching half of the wavelength of light, the diffraction limit prevents their continued miniaturization deep into the nanometer scale.

In conventional semiconductor lasers, metals are always located far from light emission regions and the cavity mode to prevent parasitic absorption by design. However, in 2003, David J. Bergman and Mark I. Stockman proposed the concept of a spaser as an amplifier of localized surface plasmons oscillating in metal, which has recently been generalized to include surface plasmon polariton amplifiers[5]. To date, there are numerous reports on laser construction based on a metallic cavity[6–28] and collectively oscillating metallic cavities[29–33]. However, immediately following the first plasmonic laser demonstrations[6–8], a debate about whether metals could really enhance the performance of lasers in general began[2,34–42]. Essentially, there is always a trade-off between field confinement and metallic absorption; plasmonics allows lasers to be made smaller but does this require additional energy to operate them? In 2014, Khurgin et al. raised the concern of whether we should use metallic cavities to construct lasers at all[38]. In particular, the role of the Purcell effect, which accompanies field localization[43], has fueled this debate. First, the Purcell effect increases the fraction, $\beta$, of excited carriers radiating into the desired cavity mode, which is known to lead to a lower laser threshold. However, accelerated spontaneous emission also consumes excited carriers faster, thus making population inversion for gain more difficult, raising the threshold[35–38,41].

Since both metal confinement and Purcell effect have arguably both positive and negative influences on nanolaser performance, fundamental questions emerge concerning the advantages of metal confinement and feedback strategies in laser technology: Are plasmonic nanolasers intrinsically high threshold due to the parasitic metal loss? And, are there situations where plasmonic nanolasers surpass the performance of photonic nanolasers?

To address these questions, in this article, we report room temperature plasmonic nanolasers with extremely low threshold on the order of 10 kW cm$^{-2}$ corresponding to a pump density in the range of modern laser diodes. We further measure 170 plasmonic and photonic nanolasers in total, each with the same gain material and cavity feedback mechanism. We systematically study their key parameters, including physical size, threshold, power consumption, and lifetime and analyze these to determine a set of laws that show how these parameters scale against each other. These scaling laws suggest that plasmonic lasers can be more compact, faster with lower power consumption than photonic lasers when the cavity size approaches or surpasses the diffraction limit. Our study also sheds light on the debate over threshold management in plasmonic lasers by the Purcell effect. While the general trend of higher threshold for higher Purcell effect is observed for both plasmonic and photonic lasers, plasmonic lasers have significantly reduced thresholds compared to photonic devices for the same recombination lifetime when the cavity size approaches or surpasses the diffraction limit. This suggests that both sides of this long-standing debate are valid with the actual physics being far from trivial.

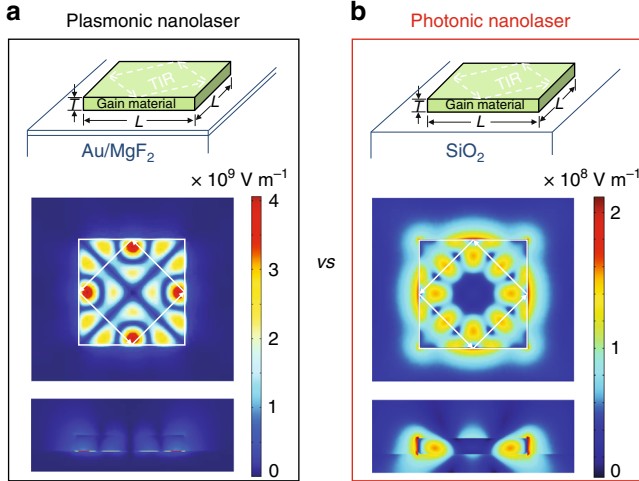

**Fig. 1** Schematic of plasmonic and photonic lasers and their cavity modes. **a** Top: schematic of the plasmonic nanolaser devices consisting of a nanosquare gain material on top of metal separated by a few nanometers of dielectric. Bottom: top and side views of electric field (|**E**|) profiles of a cavity mode in a 700 × 700 × 100 nm plasmonic cavity. **b** Top: schematic of the photonic nanolaser devices consisting of a nanosquare gain material on top of dielectric. Bottom: top and side views of electric field (|**E**|) profiles of a cavity mode in a 700 × 700 × 100 nm photonic cavity. In both panels, L and T are the length and thickness of the nanosquare, respectively, and TIR represents total internal reflection

## Results

**Device morphology.** Figure 1a shows a schematic of the plasmonic nanolaser devices. Here, light emission and gain are supplied by monocrystalline II–VI cadmium selenide nanosquares grown by chemical vapor deposition (see Methods). The nanosquares are dry transferred to gold/magnesium fluoride (200 nm/ 5 nm) substrates to form a metal (gold)–insulator (magnesium fluoride)–semiconductor (cadmium selenide) gap surface plasmon mode localized in the proximity to the insulator layer (bottom panel of Fig. 1a) (see Methods). Nanosquares with variable thickness (T) from ~50 to 1000 nm and length (L) from ~0.8 to 6 μm were studied. The morphology of all tested plasmonic and photonic nanolasers is characterized by atomic force microscopy (AFM) (Supplementary Fig. 1). In this study, we also directly compared our plasmonics lasers with photonic lasers, formed by cadmium selenide nanosquares with a similar range of T and L being placed on silicon dioxide substrates (Fig. 1b). Silicon dioxide substrates are chosen here for the low refractive index which gives better optical confinement in gain material cadmium selenide. As shown in Fig. 1, the cavity feedback is supplied by total internal reflection at the four boundaries of a nanosquare in both plasmonic and photonic lasers[11,16,44,45].

**Threshold minimization of plasmonic nanolasers.** The optical properties of cadmium selenide gain and gold plasmonic materials are optimized to achieve a low threshold when operated at room temperature. The cadmium selenide nanosquares are monocrystalline with smooth surfaces (Supplementary Fig. 2). The internal quantum efficiencies of the majority of measured plasmonic and photonic nanolasers are close to unity due to the high quality of the as-synthesized nanosquares and the Purcell enhancement accelerated radiation emission rate (Supplementary Note 1 and Supplementary Fig. 3). The gold thin film has polycrystalline structure with grain size larger than 50 nm and a high figure of merit $\left(\frac{-\text{Re}[\varepsilon_m]}{\text{Im}[\varepsilon_m]}\right)$ of 16, where $\varepsilon_m$ is the complex permittivity of gold (Supplementary Fig. 4).

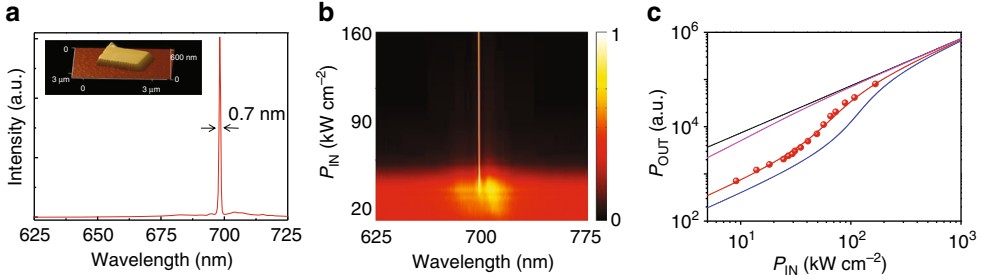

**Fig. 2** Room temperature ultralow threshold plasmonic nanolaser. **a** Laser spectrum of a plasmonic nanolaser with a threshold of ~32 kW cm$^{-2}$. Inset: atomic force microscope image of the device. **b** Spectra normalized to peak value vs. pump power highlighting the emergence of a dominant laser mode with reduced linewidth over the spontaneous emission. **c** Light–light curve of the plasmonic nanolaser (dots show data values). The spontaneous emission coupling ($\beta$) factor of the nanolaser is about 0.09 (red line). Black, pink, and blue lines are reference light–light curves corresponding to $\beta = 1$, $\beta = 0.5$, $\beta = 0.05$

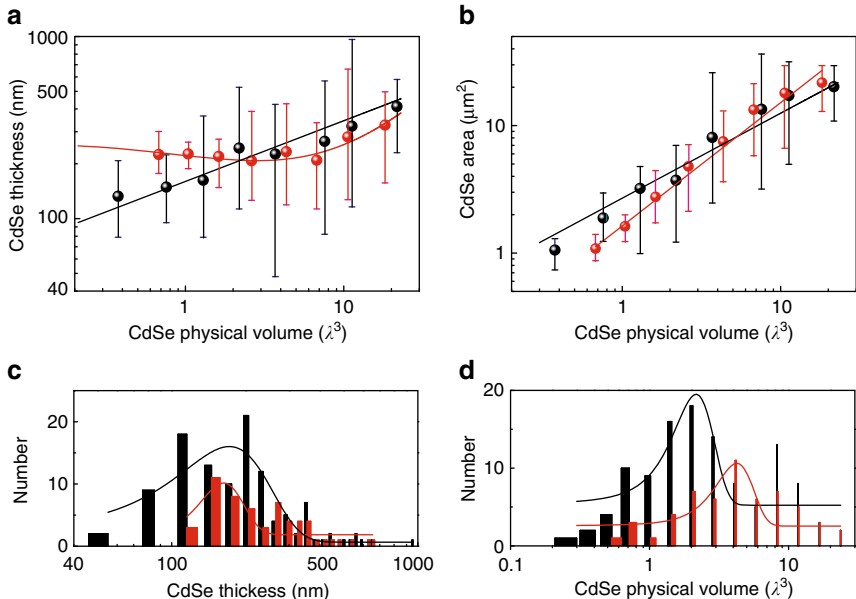

**Fig. 3** Scaling laws of physical volume vs. thickness. **a** The relationship between $T$ and $V$ for only the lasing plasmonic and photonic cavities, where $T$ and $V$ are the thickness and volume of cadmium selenide nanosquares, respectively. The relationship between $T$ and $V$ for the plasmonic nanolasers (black dots) follows the natural scaling law of $T \propto V^{1/3}$ determined by the material growth process (black line). However, for photonic devices, the scaling of $T$ shifts significantly away from the natural material scaling law when $T$ approaches the diffraction limit (red dots). The red line is a guide to the eye. Error bar shows the thickness range for a certain number of measured devices with the same volume. **b** The relationship between cadmium selenide area $A$ and $V$ for only the lasing plasmonic (black dots) and photonic cavities (red dots), where $A$ is the area of cadmium selenide nanosquares. Black line: guide to the eye of $A \propto V^{2/3}$. Red line: guide to the eye of $A \propto V$. Error bar shows the area range for a certain number of measured devices with the same volume. **c** Cadmium selenide thickness distribution histogram for plasmonic (black columns) and photonic (red columns) lasers. Lines: Gaussian distribution fittings. **d** Cadmium selenide physical volume distribution histogram for plasmonic (black columns) and photonic (red columns) lasers. Lines: Gaussian distribution fittings. In all the panels, $\lambda$ refers to 700 nm

Figure 2a shows the lasing spectrum of a typical plasmonic nanolaser with height of 140 nm, length of 2 μm, and width of 1.6 μm (inset: AFM image). The device shows pronounced single mode laser emission with a linewidth of 0.7 nm at 698 nm. The normalized spectra of the device vs. the pump power, shown in Fig. 2b, indicates a clear transition from spontaneous emission to lasing emission evidenced by the linewidth narrowing effect. The laser behavior is also evidenced by the "S" shaped light–light curve in log scale as shown in Fig. 2c where the lasing threshold is read to be about 32 kW cm$^{-2}$. The spontaneous emission coupling factor ($\beta$ factor) of the device is obtained to be about 0.09 from a fitting of the light–light curve using rate equation. The optical characterization setup is shown in Supplementary Fig. 5. We note that the devices are optically pumped by a nanosecond pump laser with pulse width comparable to the

spontaneous emission lifetime of the gain material, which helps to accumulate the excited carriers to achieve population inversion for lasing before they recombine and radiate (see Methods). The threshold values specified in this work corresponds to the peak pulse intensity of the pump laser, which is necessary to specify the threshold of a pulse pumped laser.

The low threshold achieved here has three intrinsic reasons. First, the internal quantum efficiency of cadmium selenide is optimized to approach 100%, second the polycrystalline gold film is optimized with smooth surfaces and a high material figure of merit of 16, and last total internal reflection modes are employed to lower the cavity radiation loss.

**Scaling laws of physical volume vs. thickness.** The morphologies of all 170 measured plasmonic and photonic nanolasers are

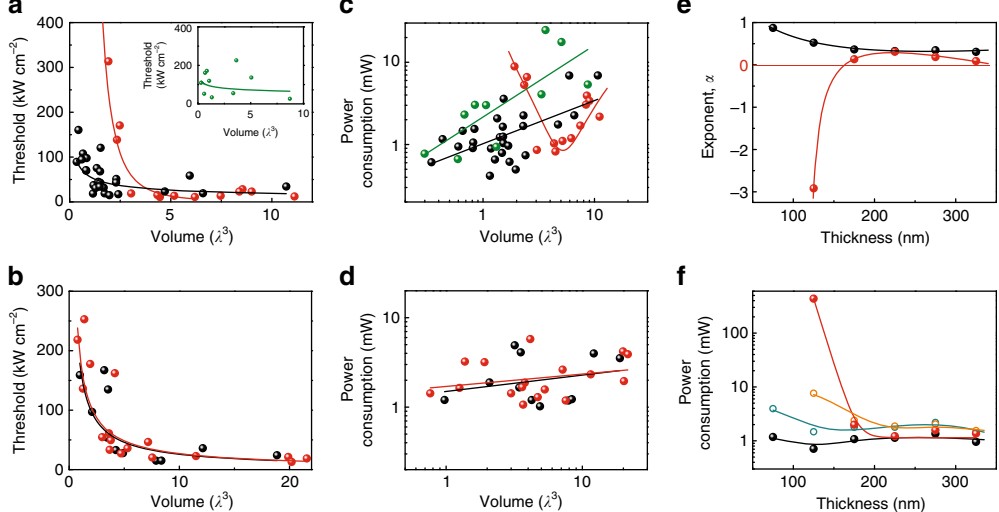

**Fig. 4** Scaling laws of threshold and power consumption vs. device sizes. **a**, **b** Laser threshold scaling as a function of device volume for plasmonic (black dots) and photonic (red dots) nanolasers with thicknesses in the range of 100 nm < T < 150 nm (**a**), and 250 nm < T < 350 nm (**b**). Inset of **a**: Laser threshold scaling as a function of device volume for plasmonic nanolasers with thicknesses in the range of T < 100 nm. **c**, **d** Power consumption scaling as a function of device volume for plasmonic nanolasers with thicknesses in the ranges of 100 nm < T < 150 nm (black dots) and T < 100 nm (olive dots) (**c**), and 250 nm < T < 350 nm (black dots) (**d**), and for photonic nanolasers with thicknesses in the ranges of 100 nm < T < 150 nm (red dots) (**c**), and 250 nm < T < 350 nm (red dots) (**d**). Lines in **a**–**d** show exponential fittings to data. **e**, **f** Quantitative analysis of power consumption scaling of plasmonic and photonic nanolasers. For each range of device thickness, a phenomenological scaling law is expressed as $P_{th}^{(power)} \propto V^{\alpha}$, where $P_{th}^{(power)}$ is the power consumption at threshold and $\alpha$ is an exponent. **e** shows $\alpha$ vs. T, highlighting the distinct scaling of power consumption of photonic (red dots) and plasmonic lasers (black dots) near the diffraction limit. **f** shows the power consumption of plasmonic and photonic nanolasers at a volume of $0.5\lambda^3$ (black dots for plasmonic nanolasers, red dots for photonic nanolasers) and $2\lambda^3$ (dark cyan circles for plasmonic nanolasers, orange circles for photonic nanolasers) for varied thickness, where lines are guides to the eye. In all the panels, $\lambda$ refers to 700 nm

characterized by atomic force microscope, which enables us to study how the physical volume of each cadmium selenide nanosquare scales as a function of thickness in each operational plasmonic and photonic laser (i.e., those that could lase). In this way, we can study how device dimension influences the ability of each device to lase. We first note that the dimensions of as-gown cadmium selenide nanosquares follow an empirical trend of $T \propto V^{1/3}$, where V is the volume. We attribute this natural scaling law to the chemical vapor deposition synthesis method, where the ratio between T and L is fairly constant and related to the material anisotropic growth rates. Second, we compared the scaling of T vs. V for those plasmonic and photonic nanolasers that successfully lased. This allowed us to determine how the cavity geometry affects the ability to achieve lasing in a very general manner.

Figure 3a shows the scaling law of physical volume vs. thickness for only the lasing plasmonic and photonic cavities. Here, the range of T values of lasing devices is shown as error bars for specific ranges of device volumes, V. For the plasmonic lasers, the relationship between T and V follows the natural scaling law of $T \propto V^{1/3}$ defined by the growth of the material. This suggests that field confinement is sustainable over the entire range of device volumes investigated, where the thinnest devices have T~50 nm. However, the scaling law for operational photonic lasers deviates significantly away from the natural scaling law of $T \propto V^{1/3}$ for V around a few $\lambda^3$ and smaller. This suggests that the optical field becomes delocalized when T approaches the diffraction limit. For small devices, photonic nanolasers only operate for shorter in-plane lengths L, but greater thickness T, which ensures a larger effective mode index for cavity feedback. Effective refractive indices and near-field distributions for plasmonic and photonic cavity modes are shown in Supplementary Figs. 6 and 7, respectively. Figure 3b shows the relationship between cadmium selenide area A and V for only the lasing plasmonic and photonic cavities. We can see that their scaling

laws of physical volume vs. area are different and match with their scaling laws of physical volume vs. thickness.

Figure 3c shows the thickness distribution histogram of all 170 devices measured. For photonic lasers, no lasing device was observed for a thickness smaller than 100 nm, because of insufficient confinement and feedback of the optical field induced by the diffraction limit. In contrast, the thickness of plasmonic lasers can be as small as about 50 nm. The same contrast also appears in the volume distribution histogram as shown in Fig. 3d.

**Scaling laws of laser threshold vs. physical volume**. We then study the scaling laws of threshold and power consumption vs. physical volume V. Figure 4a shows the laser threshold scaling law categorized by nanosquare thickness, T, for ranges near (100 nm < T < 150 nm) and below (T < 100 nm) the diffraction limit. For 100 nm < T < 150 nm, the threshold of plasmonic nanolasers increases with decreasing V with values in the range of 10–200 kW cm⁻². Meanwhile, the photonic laser threshold is in the range of 10–30 kW cm⁻² for V > ~5$\lambda^3$; however, it rises rapidly for V < ~5$\lambda^3$ and reaches 300 kW cm⁻² at V ~2$\lambda^3$, which is higher than the threshold of any other plasmonic laser with a similar V. Remarkably, when T is reduced below the diffraction limit (T < 100 nm), laser action is solely observed for plasmonic lasers. No photonic lasing is observed for over 30 devices measured in this range. When T is significantly thicker than the diffraction limit, the threshold scaling of both plasmonic and photonic lasers exhibit similar scalings, as shown in Fig. 4b. We note that the rising of the threshold with decreasing V is mainly due to greater radiative loss due to the smaller cavity in both plasmonic and photonic lasers.

Although only one dimension of the cavity is examined in the vicinity of the diffraction limit here, the physical effect of confinement loss, suggested by the scaling laws of Fig. 4a, b, is rather clear. The diffraction limit applies to any of the three

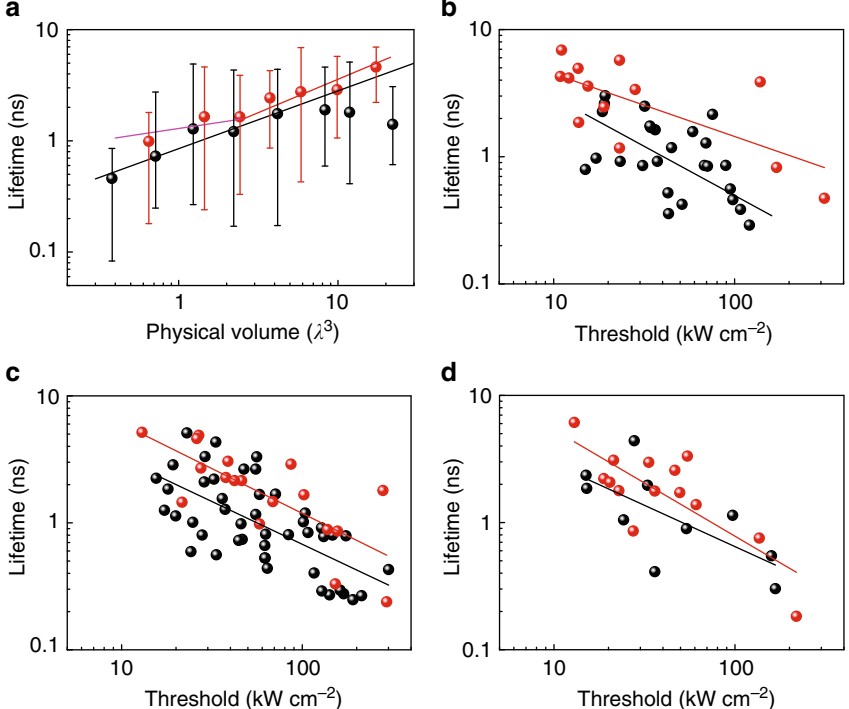

**Fig. 5** Scaling laws of emission lifetime vs. physical volume and laser threshold. **a** Scaling laws of lifetime as a function of device volume for plasmonic (black dots) and photonic (red dots) nanolasers. Error bar shows the lifetime range for a certain number of measured devices with the same volume. Lines are guides to the eye showing simulated relationship between lifetime ($\tau$) and physical volume ($V$), where black line refers to $\tau \propto V^{0.53}$, red line refers to $\tau \propto V^{0.60}$, and magenta line refers to $\tau \propto V^{0.22}$. $\lambda$ refers to 700 nm. **b**–**d** Scaling laws of emission lifetime vs. threshold of plasmonic (black dots) and photonic (red dots) nanolasers in the thicknesses ranges of 100 nm < $T$ < 150 nm (**b**), 150 nm < $T$ < 250 nm (**c**), and 250 nm < $T$ < 350 nm (**d**). Lines show exponential fittings to data by $\tau \propto P_{th}^{\alpha_1}$, where $\alpha_1$ is an exponent and $P_{th}$ is the threshold. For plasmonic lasers, the exponents are −0.77, −0.67, and −0.65 in **b**, **c** and **d**, respectively. For photonic lasers, the exponents are −0.49, −0.70, and −0.84 in **b**, **c** and **d**, respectively

dimensions of a laser cavity, because it originates from the uncertainty principle telling us that spatial localization generates a wider distribution of momenta along localized dimensions. When the thickness of a photonic nanolaser approaches the diffraction limit, the light field has to become delocalized normal to the substrate surface. Cavity mode delocalization along any dimension decreases modal overlap with the gain material and reduces the quality factor of the cavity, both of which will increase the laser threshold.

**Scaling laws of laser power consumption vs. physical volume.**
Figure 4c and d show the scaling of power consumption determined by the pump power absorbed at threshold vs. physical volume. Remarkably, while the threshold power density must increase in response to increasing cavity loss for small devices, the power required to maintain laser operation can decrease exponentially. In the case of plasmonic lasers the general trend is of a reduced power consumption with reduced gain volume for all size ranges. However, for photonic lasers with thicknesses in the range of 100 nm < $T$ < 150 nm, the power consumption rises rapidly for cavity sizes smaller than about $5\lambda^3$.

Although it is not surprising that plasmonic lasers can have a smaller physical size than photonic lasers, the contrast in laser power consumption at threshold is surprising. Indeed, the trend that a smaller laser can have a lower power consumption is one of the prime motivations for laser miniaturization. However, the scaling law of threshold power with physical volume presented in this study shows that this trend is not valid any more for photonic lasers with cavity sizes smaller than about $5\lambda^3$ when its thickness approaches the diffraction limit. Meanwhile, the trend is maintained even for the very smallest plasmonic lasers, leading

to power consumptions less than the most efficient photonic lasers near the diffraction limit. While the lowest power consumption observed from a photonic laser in this study is about 0.8 mW, the lowest power consumption of a plasmonic laser is about 0.4 mW.

Figure 4e and f show the scaling law of power consumption vs. volume in more quantitative detail, by parameterizing the data against the various thicknesses of devices. Here, we categorize plasmonic and photonic lasers with thickness from 50 to 350 nm in six ranges. For each range of $T$, a phenomenological scaling law is expressed as $P_{th}^{(power)} \propto V^{\alpha}$, where $P_{th}^{(power)}$ is the power consumption at threshold in units of mW and $\alpha$ is an exponent (Supplementary Figs. 8, 9). Figure 4e shows the relationship between exponents and $T$. We can see that while the exponents are all positive for plasmonic lasers, they become negative for thin photonic lasers supporting our observation that confinement is lost due to the diffraction limit in these devices leading to greater power consumption to maintain laser action.

Figure 4f shows the scaling of power consumption vs. device thickness for plasmonic and photonic lasers with volumes of approximately 0.5 and $2\lambda^3$ as a function of $T$. For $T$ > ~200 nm, the power consumption trends of plasmonic and photonic lasers are similar where a smaller $V$ gives a lower power consumption. However, for the thinner devices, the trends start to split and become distinct. The power consumption of photonic lasers increases rapidly with decreasing $T$, with a slope that increases for the smaller device volumes. A similar phenomenological scaling law is obtained for threshold power density as shown in Supplementary Fig. 10.

**Scaling laws of emission lifetime vs. physical volume.** Figure 5a shows the scaling law of lifetime vs. physical volume. In a cavity, the spontaneous emission lifetime ($\tau$) will be shortened by spatial and spectral localizations induced by cavity mode, known as the Purcell effect. The Purcell enhancement factor is proportional to $Q/V_m$, where $Q$ is quality factor and $V_m$ is mode volume of the cavity mode. And $\tau$ is proportional to $V_m/Q$.

We perform full electromagnetic simulations to obtain $Q$ and $V_m$ for all of the measured plasmonic and photonic lasers in order to estimate the scaling of Purcell effect with volume (Supplementary Note 2 and Supplementary Fig. 11). As the fundamental plasmonic mode (fundamental photonic mode with dominant electrical field parallel to the substrate) has the strongest in-plane field confinement and highest effective refractive index in plasmonic (photonic) cavities, we focus on its corresponding total internal reflection mode in the simulations.

Using the simulated values of $Q$ and $V_m$, we have obtained the scaling law of $V_m/Q$ for photonic and plasmonic cavities which are plotted in Fig. 5a as red, magenta and black reference lines. We can see that the experimental lifetime scaling laws match with the simulated ones. We note that the experimental data for plasmonic lasers shift away from the simulated ones for $V > \sim 5\lambda^3$, which may be due to arising of higher-order modes.

**Scaling laws of emission lifetime vs. laser threshold.** Finally, we present scaling laws of lifetime vs. laser threshold in Fig. 5b–d. Recent literature has shown that short $\tau$ in plasmonic nanolasers can lead to ultrafast device operation[19]. However, in terms of the threshold, the effect of shortening $\tau$ is not trivial. On the one hand, a shorter $\tau$ leads to a larger fraction, $\beta$, of radiation into the cavity mode, which has the effect of reducing the threshold. On the other hand, a smaller $\tau$ implies that carriers must be consumed faster making population inversion more difficult, thus raising the threshold[35–38,41].

To address this debate, we study the scaling of lasing threshold vs. carrier lifetime, $\tau$ as shown in Fig. 5b–d. The results support both of these arguments but the actual physics shows that both phenomena play a role. The overall trend in both plasmonic and photon lasers is that a decrease in lifetime corresponds to an increase in threshold. However, for these devices with thickness approaching to the diffraction limit, plasmonic lasers can have shorter lifetimes than photonic lasers for the same threshold value. Put another way, plasmonic lasers can be faster and with lower threshold simultaneously than photonic lasers when the cavity size approaches or surpasses the diffraction limit.

**Rate equation analysis on the unusual scaling laws.** The relationship between threshold and lifetime is determined by the semiconductor laser rate equations involving key laser parameters:

$$\frac{dN_2}{dt} = \eta p - R_{non}N_2 - RN_2 - \Gamma R \beta (N_2 - N_0)N_{ph} \quad (1)$$

$$\frac{dN_{ph}}{dt} = -\gamma N_{ph} + \beta RN_2 + \Gamma R \beta (N_2 - N_0)N_{ph} \quad (2)$$

Here, $N_2$ is the excited carrier population, $N_{ph}$ is the photon number of a single mode laser, $p$ is the pump rate, $N_0$ is the excited state population at transparency, $\eta$ is the conversion efficiency of pump photons into electron/hole pairs, $R_{non}$ is the non-radiative recombination rate, $R$ is the spontaneous emission rate ($\sim 1/\tau$), $\Gamma$ is the confinement factor, $\beta$ is the spontaneous emission coupling factor, $\gamma$ is the total cavity loss rate.

Under quasi-steady state conditions of the experiments, the solution to the threshold power density using the measure of the photon number transition is (Supplementary Note 3):

$$P_{th} = \frac{h\nu}{\eta} \left[ \frac{1 - \eta_i \beta}{\eta_i \Gamma \beta A} \cdot \gamma + R_{non} n_0 T + (1 - \beta) n_0 T \cdot \frac{1}{\tau} \right] \quad (3)$$

where $h\nu$ is the emitted photon energy, $\eta_i$ is the internal quantum efficiency $\left( \frac{R}{R_{non}+R} \right)$ of cadmium selenide, $n_0$ is the excited carrier density at transparency, $T$ and $A$ are the thickness and area of cadmium selenide nanosquares, respectively.

The first term of $P_{th}$ is proportional to $\gamma$ inversely proportional to $Q$ accounting for the cavity photon loss compensation. The second term is proportional to $R_{non}$, accounting for non-radiative carrier loss. The last term, which is proportional to $n_0/\tau$, accounting for the power density required to achieve carrier population inversion.

The debated role of Purcell effect on the laser threshold is clear in Eq. (3). First, the Purcell effect increases $\beta$, which reduces the first and last terms. However, accelerated spontaneous emission reduces $\tau$, making the last term larger. Because the internal quantum efficiency is close to unity for both plasmonic and photonic lasers (non-radiative emission is negligible), in the following, we focus on the first and the last terms here.

We now compare the theoretically predicted thresholds from Eq. (3) with experimental data. For each thickness range, we empirically fit the experimental data in Fig. 5b–d by $\tau \propto P_{th}^{\alpha_1}$, where $\alpha_1$ is an exponent. For plasmonic lasers, the exponents are −0.77, −0.67, and −0.65 for the thicknesses ranges of 100 nm < $T$ < 150 nm, 150 nm < $T$ < 250 nm, and 250 nm < $T$ < 350 nm, respectively. For photonic lasers, the exponents are −0.49, −0.70, and −0.84 for the thicknesses ranges of 100 nm < $T$ < 150 nm, 150 nm < $T$ < 250 nm, and 250 nm < $T$ < 350 nm, respectively.

First, we can see that for all thickness ranges of plasmonic lasers and thickness thicker than 150 nm of photonic lasers, $\alpha_1$ is close to but larger than −1, which indicates that the last term of Eq. (3) dominants over the first term. We note that, for both plasmonic and photonic lasers, the absorption from the gain material should contribute a significant portion to the laser threshold[27]. And clearly, the shortened $\tau$ induced by small cavity dominants the scaling law in this range.

Second, $\alpha_1$ for photonic lasers becomes significantly larger than that for plasmonic lasers when the thickness of cadmium selenide is between 100 and 150 nm. Here, the increasing total cavity loss rate, $\gamma$ from the first term of Eq. (3) should cause the dramatic performance degradation of photonic lasers compared to the plasmonic ones. $Q$ of a photonic cavity is very sensitive to the thickness or side length change in this range (Supplementary Fig. 12). We note that the modes of thinner photonic cavities are more delocalized compared to plasmonic cavities, which results in greater sensitivity to surface roughness and cavity irregularities. Practically, the reduced confinement factor $\Gamma$ will also play a role in the threshold increase of the photonic lasers in this thickness range.

Meanwhile, the field of the fundamental plasmonic cavity mode is well confined and their quality factors, while being low, remain inert to size variations until the thickness is just tens of nanometers (Supplementary Fig. 12). The smaller achievable plasmonic cavity size also leads to a high $\beta$, and thus these cavities can have shorter lifetimes and lower thresholds simultaneously compared to photonic lasers. We note that the above analysis are based on the quasi-steady state conditions of the experiments. A time-dependent lasing buildup process comparison between plasmonic and photonic lasers is also interesting but not yet included.

## Discussion

In conclusion, we report a room temperature plasmonic nanolaser at extremely low thresholds on the order of $10\,\mathrm{kW\,cm^{-2}}$ corresponding to a pumping density in the range of modern laser diodes. Four scaling laws of plasmonic and photonic nanolasers have been studied as a function of device volume, namely, thickness; threshold [$\mathrm{W\,cm^{-2}}$]; power consumption at threshold [mW]; and lifetime. We have also studied the scaling law of emission lifetime vs. laser threshold. In contrast to the photonic devices, plasmonic nanolasers have unusual scaling laws allowing them to be more compact and faster with lower power consumption when their cavity size approaches or surpasses the diffraction limit. These results indicate that while threshold increases are inevitable with reducing the size of plasmonic lasers, they are sufficiently gradual to allow plasmonic lasers to operate with decreasing power consumption. The origin of this effect is associated with confinement and the Purcell effect, which allows plasmonic lasers to maintain lower thresholds than photonic lasers for similar values of lifetime. Our results provide unambiguous evidence that plasmonic lasers have superior performance over photonic lasers when their dimensions are at the nanoscale, which clarifies the long-standing debate over the viability of metal confinement and feedback strategies in laser technology. The ultralow threshold of plasmonic nanolasers demonstrated here will pave the way for their practical applications in various field including low power consumption nanophotonic circuitry, super-resolution imaging, and ultrasensitive biochemical sensors.

## Methods

**Materials growth**. Cadmium selenide nanosquares were synthesized via chemical vapor deposition method[46]. Cadmium selenide (99.99%) powder was used as the source, and pieces of silicon wafer covered with 10 nm thick thermally evaporated gold catalysts were used as the substrates. Before heating, the quartz tube was cleaned by high-purity argon for 20 min to remove the oxygen. Then the furnace was rapidly heated to 700 °C, the growth duration was set as 30 min under a high-purity argon flow with flow-rate of 100 standard-state cubic centimeter per minute.

**Device fabrication**. For plasmonic nanolasers, cadmium selenide nanosquares grown on silicon substrates were dry transferred via a face-to-face sliding method onto the magnesium fluoride/gold (5 nm/200 nm) thin film structure which was deposited by electron-beam evaporation. For photonic nanolasers, 600 nm thick silicon dioxide on silicon wafers were used as substrates.

**Crystalline characterization**. A high-resolution transmission electron microscope (TECNAI F30) was used for charactering crystalline structures of cadmium selenide nanosquares and gold substrates.

**Gold permittivity characterization**. The permittivity of the gold substrate was measured by an ellipsometer (HORIBA UVISEL: 200–2000 nm).

**Physical volume characterization**. The physical volumes of the nanolasers were measured by atomic force microscope (BRUKER Dimension Icon).

**Threshold characterization**. The nanolasers were optically pumped by a nanosecond pump laser ($\lambda_{\mathrm{pump}} = 532$ nm, repetition rate: 1 kHz, pulse length: 4.5 ns). A $20\times$ objective lens (NA = 0.4) was used to focus the beam to a ~20 μm diameter spot on the sample surface to pump the nanolasers. The emission from the nanolasers was collected in reflection by the same objective and analyzed by a spectrometer (ANDOR SR-500i-B2-R).

The power consumption at threshold $P_{\mathrm{th}}^{(\mathrm{power})}$ is calculated as $P_{\mathrm{th}}^{(\mathrm{power})} = P_{\mathrm{th}} \times A \times \alpha_{\mathrm{T}}$, where $P_{\mathrm{th}}$ is the power density at threshold, $A$ is the nanosquare area, $\alpha_{\mathrm{T}}$ is a ratio of a nanosquare absorbed power over incident power related to the interface reflections and materials absorption coefficient. $\alpha_{\mathrm{T}}$ is calculated by:

$$\alpha_{\mathrm{T}} = 1 - R_0 - \left[ R_1(1-R_0)e^{-\alpha_{\mathrm{abs}}T} + (1-R_1) \right] \frac{(1-R_0)e^{-\alpha_{\mathrm{abs}}T}}{1 - R_1 R_0 e^{-2\alpha_{\mathrm{abs}}T}}$$

where $\alpha_{\mathrm{abs}}$ is the absorption coefficient of cadmium selenide at the wavelength of pump light, $R_0$ and $R_1$ are the reflection coefficient at the interface of cadmium selenide/air and cadmium selenide/substrate, respectively, and $T$ is the thickness of cadmium selenide nanosquare.

**Lifetime characterization**. An exponential model is used to fit time-correlated single photon counting spontaneous emission histograms measured by a PicoHarp 300 to obtain the lifetime. The formula of the fitting is as follows: $I(t) = \int_{-\infty}^{t} \mathrm{IRF}(t')Ae^{-\frac{t-t'}{\tau}}dt'$, where $I(t)$ is the time-resolve spontaneous emission histogram and $A$ is the amplitude. $\mathrm{IRF}(t')$ is the instrument response function and $\tau$ is the lifetime.

**Data availability**. The data that support the findings of this study are available from the authors on reasonable request.

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

## Acknowledgements

This work was supported by the National Natural Science Foundation of China (nos. 11574012, 11774014, 61521004, and 11474007), the Youth 1000 Talent Plan Fund, and the National Basic Research Program of China (nos. 2013CB921901).

## Author contributions

R.-M.M. conceived and provided guidance to the work. S.W. and B.L. carried out the experiments. X.-Y.W., S.W., and H.-Z.C. conducted theoretical simulations. Y.-L.W. and L.D. synthesized cadmium selenide nanosquares. R.-M.M. and S.W. wrote the manuscript. All authors discussed the results and revised the manuscript.

## Additional information

**Competing interests:** The authors declare no competing financial interests.

