## [Peer Review File · Nature Communications]

Reviewers' comments:

Reviewer #1 (Remarks to the Author):

Review of the manuscript "Unusual Scaling Laws for Plasmonic Lasers beyond Diffraction Limit" by S. Wang et al.

The authors of the manuscript compared 170 miniature (above the diffraction limit and below the diffraction limit) plasmonic and photonic lasers based on CdSe squares grown by the chemical-vapor-deposition method – the experimental work of a gigantic proportion. In photonic lasers, CdSe squares have been placed on top of SiO₂, while in plasmonic lasers – on top of Au with the MgF₂ spacer. The stimulated emission threshold in plasmonic lasers was demonstrated to be as low as ~10 KW/cm² (the remarkably low value !). The dependences of the stimulated emission threshold, power consumption (at the threshold), and the spontaneous emission life-time have been experimentally studied as a function of the height and volume of the CdSe squares. The two major questions have been addressed: (1) whether plasmonic lasers (lossy by their nature) can out-perform photonic lasers and (2) whether and how the Purcell enhancement of spontaneous emission affects the laser performance.

The answer to the first question, although not highly surprising, is encouraging to the plasmonic community: in sub-diffraction plasmonic lasers, both the threshold and the power consumption are smaller than those in the photonic lasers of the same size. The direct experimental proof of this fact is important, as it is going to settle down the remaining ongoing debates.

Figure 4c shows the reduction of the power consumption (in the plasmonic laser) with reduction of the laser volume. This experimental result seems to be in disagreement with recent claims of J. Khurgin, who has predicted that in sub-diffraction plasmonic lasers, the power consumption should saturate with the further size reduction. The authors are encouraged to comment on this discrepancy.

The answer to the second question is obvious to many researchers working in the field of laser physics. However, the fact that the same question is asked again and again justifies addressing it one more time. It is known from the dawn of quantum electronics that the spontaneous emission is an adversary of the stimulated emission, as both of them compete for the same excited state population. In particular, the fact that the ratio between the Einstein's coefficients A and B is proportional to the cubic power of frequency, hinders short-wavelength lasers. This is true for all laser sizes, including miniature lasers and spasers.

However, there is a misconception exists that by increasing the rate of spontaneous emission (Purcell enhancement) and by radiating this emission into the laser mode (high beta factor), one can reduce the lasing threshold and, eventually, create a threshold-less laser. In reality, this effect straightens the input/output curve and reduces its bent. However, it does not reduce the threshold, defined e.g. as the point at which the photon statistics does change. The only possible change of the threshold is its increase due to competition of the spontaneous and the stimulated emission channels for the excited state population. The authors' answer to this question is physically correct, however, it would benefit from more clarity.

The reviewer recommends to publish the paper after the clarity is improved and the comments below are adequately addressed.

1. The main body of the manuscript should be "self-sufficient": all its main points should be understandable without looking into the Supplementary Information section.
2. Can anything be learned from knowing the dimensions of the samples, which did not lase?
3. Page 7: "Figure 2 c and d shows the scaling of power ..."
 - The reference to the figure seems to be wrong.
4. Page 8: "For each range of T, a phenomenological scaling law is expressed as $P_{th} \propto \Omega V a$ ".
 - Why this fitting formula was chosen?
5. Page 9: "We note that such an increase of tau for photonic lasers with decreasing cavity volume is also observed in simulation results, shown in Supplementary section 13."
 - Why does this happen?
6. Page 9: "Four scaling laws of threshold, power consumption, physical volume and lifetime are experimentally revealed"

- Only three are listed.

7. Page 10: "The origin of this effect seems to be associated with confinement and the Purcell effect, which allows plasmonic lasers to maintain lower thresholds than photonic lasers for similar values of lifetime."

- More explanation/clarity is needed.

8. What are the slopes of the curves in Figs. 4a,b,c,d, 5a,5b? Can anything be learned from these slopes?

Reviewer #2 (Remarks to the Author):

This manuscript describes the scaling laws of plasmonic lasers with physical sizes close to the diffraction limit. Impressively, the authors studied > 170 samples, and a side-by-side comparison between plasmonic and photonic lasers regarding lasing threshold, power consumption, and lifetime was carried out. Empirically, 53 was found to be the critical physical size to distinguish the performance between plasmonic and photonic nanolasers.

In general, the authors should provide more physical interpretations of their results to enhance their experimental data analysis. For example, the intrinsic cavity modes and near-field distributions in both plasmonic and photonic lasers need to be clarified. The authors should provide more theoretical analysis on why 53 is the critical volume number. The mode delocalization with decreased cavity size need to be further verified by near-field distribution results (page 7, Figure S19).

Additionally, the lateral dimension and disk thicknesses were not differentiated in their comparison among different physical volumes. For example, the crystal thickness of nano-squares with the same volume ranged from 70 nm to 400 nm (Figure 3a). Especially for thick nano-disks, higher order waveguide mode in the vertical direction can also form within the cavity and provide additional optical feedback for lasing action.

Also, the authors did not provide the rationale underlying their design considerations of photonic lasers. For example, why do they use silica as the substrate? Will the threshold and energy consumption behavior change if other substrates such as sapphire are used, which has higher thermal conductivity and could cool down the device? Also, the refractive index of different substrates will influence the field confinement and device performance of photonic nanolasers.

Other comments / questions include:

1. The authors should specify that the record low lasing thresholds are those compared to other hybrid plasmonic waveguide lasers. For plasmonic lasers based on nanoparticle arrays [Nature Nanotechnology 8, 506–511 (2013).], lasing thresholds are around 0.2 W cm^{-2} (0.2 mJ cm^{-2} with 1 kHz operation), which is lower than the number reported (10 KW cm^{-2}) in this manuscript. These two different systems should be differentiated in the background section.

2. The authors should explain more why the record low lasing threshold was achieved. Is it because of the polycrystalline Au film with smoother surfaces, or improved quantum efficiency of CdSe? Such information will be beneficial for the nanophotonics community.

3. More systematic studies are needed on the near-field distribution of the intrinsic cavity modes. The authors claimed that taller nano-squares can support stronger photonic resonances, but only a few cavity sizes were analyzed, and the near-field distribution of taller disks is missing. In plasmonic nanolasers, larger physical sizes can also lead to higher-order modes, which is possibly the reason for decreased lasing threshold at large physical volumes (Figure 5a).

4. The authors should justify their mode volume calculations considering the intrinsic divergent nature of plasmonic modes. Related theoretical analysis can be found in ACS Photonics 1, 2-10 (2014) and Physical Review Letters 110, 237401 (2013).

5. Can the authors perform more theoretical analysis to predict the lasing threshold for plasmonic and photonic cases? Their current theoretical analysis focused on quality factors and mode volumes of the cold cavity. More gain-loss analysis by incorporating gain within the modeling is essential to verify the trends observed in experiments and to reveal the mechanism behind the unusual scaling laws.

6. In general, the figures can be improved for better visualization. For example, the pink curves and text in Figure 4 were hard to resolve, and the figure resolutions (especially those in supplementary information) should be higher.

7. The fitting curves deviated much from experimental data in Figures 4a-d. Can the authors explain the origin of big fluctuations in their measured data?

Overall, this paper described unusual scaling laws of plasmonic nanolasers below the diffraction limit. This systematic study and comparison of photonic and plasmonic lasers will be highly appreciated by the community. However, more validation and interpretation are needed to understand their designs of photonic lasers and the comparison among different physical volumes. Once these major concerns are addressed, the paper will provide important and critical insight for this increasingly interesting topic.

Response Letter to Reviewer 1

We thank reviewer 1 for the encouraging remarks of our work including comments such as “the experimental work of a gigantic proportion...The two major questions have been addressed...The direct experimental proof of this fact is important, as it is going to settle down the remaining ongoing debates...”, and “The reviewer recommends to publish the paper after the clarity is improved and the comments below are adequately addressed.”. By carefully conducting new simulation and data analysis, we are now able to address all of the reviewer’s comments. We believe that these questions have helped us to improve significantly the quality and presentation of the manuscript.

Reviewer’s general comments:

Figure 4c shows the reduction of the power consumption (in the plasmonic laser) with reduction of the laser volume. This experimental result seems to be in disagreement with recent claims of J. Khurgin, who has predicted that in sub-diffraction plasmonic lasers, the power consumption should saturate with the further size reduction. The authors are encouraged to comment on this discrepancy.

Our responses:

We thank the reviewer for his/her comment. In general, the threshold of a laser should decrease with its size reduction. The main reasons for such a trend are two folds: (1) the size reduction of a laser will decrease the number of modes inside the cavity and thus increase the spontaneous emission coupling factor (β), which will lower the threshold; (2) the size reduction of a laser will decrease the total number of excited carriers necessary for population inversion, which will naturally lower the threshold.

It is our understanding that, the statement, “*in sub-diffraction plasmonic lasers, the power consumption should saturate with the further size reduction*” should be only valid for extremely small cavity sizes, where the physical volume of the cavity should be significantly smaller than $(\lambda/2n)^3$ and the β factor tends to 1. In such a situation, the

power consumption is determined dominantly by the cavity loss rate, and should saturate with further size reduction. However, as shown by the experimental result here, even photonic lasers reach a similar limit with a physical size on the order of λ^3 , which is about two orders of magnitude larger than that of plasmonic ones. It is clear that the metal can significantly enhance laser performance at nanoscale dimensions.

We have added the following sentence to the revised manuscript to clarify this:

“We note that scaling laws for plasmonic lasers with even smaller cavity sizes would require further study. At extremely small cavity size region, where the β factor tends to 1, light-light curve will straighten to be a straight line known as a ‘thresholdless’ behavior, and a more detailed theory is needed to define a laser’s threshold, for example using phase space methods or photon statistics. [46, 47]. At such a situation, the power consumption should be determined dominantly by cavity loss rate.”

Reviewer’s general comments:

However, there is a misconception exists that by increasing the rate of spontaneous emission (Purcell enhancement) and by radiating this emission into the laser mode (high beta factor), one can reduce the lasing threshold and, eventually, create a threshold-less laser. In reality, this effect straightens the input/output curve and reduces its bent. However, it does not reduce the threshold, defined e.g. as the point at which the photon statistics does change. The only possible change of the threshold is its increase due to competition of the spontaneous and the stimulated emission channels for the excited state population.

The authors’ answer to this question is physically correct, however, it would benefit from more clarity.

Our responses:

We agree with the reviewer that it is impossible to achieve zero threshold through the Purcell effect and corresponding increase in β factor (the ratio of spontaneous emission channeled to lasing mode over all spontaneous emission). In a review paper that we

published in 2013, we stated that, “the term ‘thresholdless’ should be understood to mean ‘lack of threshold behavior’, as opposed to ‘an onset of laser action for infinitely small pump rates’” (*Laser & Photonics Reviews*, **7**, 1-21, (2013)).

The threshold of lasers relies on a series of interconnected parameters, including β factor, spontaneous emission rate, stimulated emission rate, non-radiative emission rate, cavity loss rate, and optical confinement factor inside gain material. A high β factor can only reduce the onset of threshold to a finite level; once the beta factor approaches unity, the threshold of a laser does not disappear, so the *straightened* input/output curve will not be valid to define threshold any more. As pointed out by the reviewer, other methods, such as *photon statistics* change should be used to define the threshold of a laser in this circumstance [PRL 98, 043906 (2007)].

To address the reviewer’s request for us to improve clarity, we have added the following sentence and citations to the revised manuscript.

“At extremely small cavity size region, where the β factor equals to 1, the light-light curve will straighten to give the impression of ‘thresholdless’ behavior. In such a situation, a more detailed theory is needed to define a laser’s threshold, for example using phase space methods or photon statistics. [39, 40].”

Reviewer’s comments:

1. The main body of the manuscript should be “self-sufficient”: all its main points should be understandable without looking into the Supplementary Information section.

Our responses:

The manuscript is self-contained and consistent. The supplementary information file serves to provide background information; however, we admit that it is quite extensive and gives the impression that vital information might be contained there. Currently, the format is not consistent with Nature Communications guidelines on supplementary information and so we have modified this document. Now the information is presented as specific supplementary figures and notes referred to specifically in the main text. We

believe this condenses the information in the SI file and shows that the manuscript is self-contained.

Reviewer's comments:

2. *Can anything be learned from knowing the dimensions of the samples, which did not lase?*

Our response:

Yes – this is an important aspect of this work. We thank the reviewer for pressing this point, which forces us to improve the clarity of this part in the manuscript. On page 5, the original manuscript reads:

“The morphologies of all 170 measured plasmonic and photonic nanolasers are characterized by AFM, which enables us to study how the physical volume of each CdSe nanosquare scales as a function of thickness in each operational plasmonic and photonic laser (i.e. those that could lase).”

This sentence describes how we will analyze the effect of device dimensions on the laser response. To make this clearer to readers, we have added another sentence as follows:

“In this way, we can study how device dimension influences the ability of each device to lase.”

The manuscript continues:

“We first note that CdSe nanosquares used to construct nanolasers follow a natural trend of $T \sim V^{1/3}$, where T is nanosquare thickness and L is the in-plane length. We attribute this natural scaling law to the chemical-vapor-deposition synthesis method, where the ratio between L and T is fairly constant and related to the material anisotropic growth rates.”

These sentences relay the point that there is a statistical relationship between nanosquare thickness and volume as a consequence of the fabrication process. This paragraph has now been modified, as follows, to make this clearer.

“We first note that the dimensions of as-grown CdSe nanosquares follow an empirical trend of $T \sim V^{1/3}$, where T is thickness and V is the volume. We attribute this natural scaling law to the chemical-vapor-deposition synthesis method, where the ratio between L and V is fairly constant and related to the material anisotropic growth rates.”

The discussion culminates in some general remarks about the effect of device dimension on laser operation in the photonic and plasmonic devices:

“For the plasmonic lasers, the relationship between T and V for the plasmonic nanolasers follows the same natural scaling law of $T \propto V^{1/3}$ defined by the material’s growth. This suggests that field confinement is sustainable over the entire range of device volumes investigated, where the thinnest devices have $T \sim 50$ nm. However, the scaling law for operational photonic lasers deviates significantly away from the natural scaling law of $T \propto V^{1/3}$ for V around a few λ^3 and smaller. This suggests that the optical field becomes delocalized when T approaches the diffraction limit.”

We believe that the modified manuscript now clearly details the influence of device dimension on the ability for the plasmonic and photonic devices to lase.

Reviewer’s comments:

Page 7: “Figure 2 c and d shows the scaling of power ...”

- The reference to the figure seems to be wrong.

Our response:

We thank the reviewer for pointing this out. It should be Figure 4 c and d. This has been changed in the manuscript accordingly.

Reviewer’s comments:

4. Page 8: “For each range of T , a phenomenological scaling law is expressed as $P_{th} \propto \Omega Va$ ”.

- Why this fitting formula was chosen?

Our response:

The fitting formula was chosen to be an empirical power law as there are complicated relationships between the various parameters that govern threshold. The threshold of a laser relies on a number of key parameters, including conversion efficiency of pump photons into electron/hole pairs, nonradiative recombination rate, spontaneous emission rate, confinement factor, spontaneous emission coupling factor and cavity mode loss rate. Apart from the conversion efficiency of pump photons into electron/hole pairs and nonradiative recombination rate, all other parameters are functions of the physical volume of the cavity with a given geometry configuration. Thus determining the specific formula of threshold versus volume is intractable. Nonetheless, the relationship between laser threshold and physical volume is extremely important, being one of the prime motivations for laser miniaturization.

A more detailed discussion on the relationship between laser threshold and physical volume based on laser rate equation analysis follows. We kindly refer the reviewer to see our responses to his/her comments 7 and 8.

Reviewer’s comments:

5. Page 9: “We note that such an increase of τ for photonic lasers with decreasing cavity volume is also observed in simulation results, shown in Supplementary section 13.”

- Why does this happen?

Our response:

The spontaneous emission lifetime (τ) of an emitter into a cavity mode should be enhanced by spatial and spectral confinements of the cavity mode, known as the Purcell

effect. Spatial and spectral confinements can be quantitatively characterized by cavity mode volume (V_m) and quality factor (Q). When an emitter and a cavity is on resonance, τ should be inversely proportional to Q/V_m .

Here the increase of τ for photonic lasers with decreasing cavity volume is due to the lack of spatial and spectral confinements when the cavity size approaches the diffraction limit. In Fig. S27, S28 of the original SOM, the simulation results show that, for photonic cavities, the quality factor will decrease with the decrease of the cavity volume, while the mode volume will increase with the decrease the cavity volume when any dimension of the cavity approaches the diffraction limit. Consequently, Q/V_m will decrease with the decrease of cavity volume, which results in an increase of τ ($\sim V_m/Q$) with decreasing cavity volume for photonic lasers.

Reviewer’s comments:

6. Page 9: “Four scaling laws of threshold, power consumption, physical volume and lifetime are experimentally revealed”

- Only three are listed.

Our response:

The four scaling laws are:

1. Device thickness versus physical volume
2. Threshold [Wcm^{-2}] versus device volume
3. Power consumption at threshold [mW] versus device volume
4. Lifetime versus device volume

To make this clearer, we have re-written the sentence on page 9 as follows:

“Four scaling laws of plasmonic and photonic nanolasers have been studied as a function of device volume, namely: threshold [Wcm^{-2}]; power consumption at threshold [mW]; thickness; and lifetime.”

Reviewer's comments:

7. Page 10: “The origin of this effect seems to be associated with confinement and the Purcell effect, which allows plasmonic lasers to maintain lower thresholds than photonic lasers for similar values of lifetime.”

- More explanation/clarity is needed.

Our response:

This is an important aspect of this work. We thank the reviewer for asking this question, which helps us to improve the clarity of this part in the manuscript.

To clarify why plasmonic laser can simultaneously have shorter lifetime and lower threshold, we conducted a steady state rate equation analysis to explore the relationship between threshold and lifetime as well as other key laser parameters rigorously. We also conducted further data analysis on Fig. 5b from the original manuscript. The following is a discussion of the new materials for the benefit of the reviewer. This material now appears in the manuscript and supplementary information files.

We first start with semiconductor rate equations that follow:

$$\frac{dN_2}{dt} = \eta p - R_{\text{non}}N_2 - RN_2 - \Gamma R \beta(N_2 - N_0)N_{\text{ph}} \quad (1)$$

$$\frac{dN_{\text{ph}}}{dt} = -\gamma N_{\text{ph}} + \beta RN_2 + \Gamma R \beta(N_2 - N_0)N_{\text{ph}} \quad (2)$$

Here, N_2 is the excited carrier population, N_{ph} is the photon number of a single laser, p is the pump rate, N_0 is the excited state population at transparency, η is the conversion efficiency of pump photons into electron/hole pairs, R_{non} is the nonradiative recombination rate, R is the spontaneous emission rate, Γ is the confinement factor, β is the spontaneous emission coupling factor, γ is the total cavity loss rate.

The first equation describes the rate of change of the carrier population, while the second equation describes the rate of change of the photon number. In the quasi steady

state conditions considered in our experiments, $\frac{dN_2}{dt} = 0$ and $\frac{dN_{ph}}{dt} = 0$, the photon number N_{ph} can be described by the quadratic equation:

$$\gamma N_{ph}^2 - [\eta p - \frac{R_{non} + R}{\Gamma R \beta} \gamma - (R_1 + R)N_0 + \beta R N_0] N_{ph} - \frac{\eta p}{\Gamma} = 0 \quad (3)$$

The solution of the above equation has two linear trends at large and small pump rates respectively, which define the conventional kink in the light-light curve. We define the threshold at the value of power density where these linear trends intersect. The pump power density is related to pump rate by $P = R_p h\nu / A$, where $h\nu$ is the photon energy and A is the device area. The threshold P_{th} is thus,

$$P_{th} = \frac{h\nu}{\eta} \left[\frac{1 - \eta_i \beta}{\eta_i \Gamma \beta A} \cdot \gamma + R_{non} n_0 T + (1 - \beta) n_0 T \cdot \frac{1}{\tau} \right] \quad (4)$$

Here, $\eta_i = \frac{R}{R_{non} + R} \approx 1$ is the internal quantum efficiency of the gain material, so that $R_{non} \ll R$. $\tau \approx \frac{1}{R}$ is the spontaneous emission lifetime. $n_0 = \frac{N_0}{V}$ is the excited state density at transparency where V is the volume of the CdSe. $T = V/A$ is the thickness of the CdSe nanosquare.

The first term of P_{th} is proportional to γ inversely proportional to Q accounting for the cavity photon loss compensation. The second term is proportional to R_{non} , accounting for nonradiative carrier loss. The last term, which is proportional to n_0/τ , accounting for the power density required to achieve carrier population inversion.

The debated role of Purcell effect on the laser threshold is clear in equation (3). Firstly, the Purcell effect increases β , which reduces the first and last terms. However, accelerated spontaneous emission reduces τ , making the last term larger. Because the internal quantum efficiency is close to unity for both plasmonic and photonic lasers (nonradiative emission is negligible), in the following, we focus on the first and the last terms here.

We can see that, for a giving thickness T , P_{th} should be inversely proportional to τ providing the last term dominants the P_{th} and β is significantly small then unit.

To clarify the origin of plasmonic lasers having simultaneously shorter lifetime and lower threshold, we further analyzed the relationship between lifetime and threshold by categorizing the data in Fig. 5b by nanosquare thickness. As shown in Fig. R1, the difference between plasmonic and photonic lasers becomes more distinct for thinner devices.

Figure R1: (Added as Fig. 5b-d in the revised manuscript) Scaling of lifetime with threshold for plasmonic (black dots) and photonic (red dots) lasers categorized by CdSe nanosquare thickness as marked in the panels. Solid lines are fitted lines.

We now compare the theoretical predicted thresholds from equation (4) with experimental data. For each thickness range, we empirically fit the experiment data in Fig. R1 by $\tau \propto P_{th}^{\alpha_1}$, where α_1 is an exponent and is depicted in Fig. R1 .

We can see that, (I) for all of thickness ranges of plasmonic lasers and thickness thicker than 150 nm of photonic lasers, α_1 is close to but larger than -1, which indicates that the last term of equation (3) dominants over the first term. We note that, for both plasmonic and photonic lasers, the absorption from the gain material should contribute a significant portion to the laser threshold. And clearly, the shortened τ induced by small cavity dominants the scaling law in this range. (II) α_1 for photonic lasers become significantly larger than that for plasmonic lasers when the thickness of CdSe is between 100-150 nm.

In order to gain further insight into this phenomenon, we must also analyze the relationship between threshold and physical volume, as shown in Fig. 4 a and b in the

original manuscript (and Fig. R3 replotted here in a linear scale). Importantly, the threshold of photonic lasers will dramatically increase with the decrease of physical volume when the thickness of CdSe is between 100-150 nm.

Figure R2: (New Fig. 4a and b in the revised manuscript) Scaling of threshold with physical volume in linear scale for plasmonic (black dots) and photonic (red dots) lasers categorized by CdSe nanosquare thickness as marked in the panels.

From Fig. R1 and R2, we can see a general trend for both plasmonic and photonic lasers that a smaller physical volume and a shorter lifetime will result in a higher threshold. But for the lasers with thicknesses approaching the diffraction limit, plasmonic lasers clearly have simultaneously shorter lifetime and lower threshold. Among the parameters in the first and last terms of equation (1), the total cavity loss rate, γ should be the fact that causes such a dramatic performance degradation of photonic lasers compared to plasmonic ones. The cavity loss γ can be expressed as $\frac{\omega}{Q}$, where Q is the quality factor and ω is the resonant frequency of the cavity. For a giving thickness range, all other parameters should not change dramatically with the reduction of the side length and consequently physical volume to a few λ^3 . However, Q of a photonic cavity is very sensitive to the thickness or side length change in this range as shown in Fig. R3.

Practically, the decreased confinement factor Γ in the thickness range approaching the diffraction limit should be another factor for the poor performance of photonic lasers. As shown in Fig. R4, for thinner devices, photonic cavity has a more delocalized mode

compared to the plasmonic cavity, which results in a quality factor that is more sensitive to the surface roughness and cavity irregularities.

For plasmonic lasers, the field is well confined (Fig. R4) and their quality factor is inert to the size change till the thickness approaches tens of nanometer (Fig. R3). And thus they can have higher Purcell factor (shorter lifetime) and lower threshold simultaneously comparing to photonic lasers at small size.

Figure R3: (Added as Figure S12 in Supp. Inf. File.) Scaling of quality factor Q with physical volume for plasmonic and photonic laser cavity modes. For each case, the quality factor Q is calculated for a CdSe nanosquare with thickness of 60 nm, 100 nm, 150 nm and 400 nm respectively. The dotted lines in both figures indicate the same Q value. Here, all values of Q are for total internal reflection modes of TM_{00} mode and TE_{00} mode, which are with strongest field confinement and highest effective refractive index in plasmonic and photonic cavities respectively.

We have made the following changes in the revised manuscript to clarify this:

- (1) We have added a section of “Rate equation analysis on the unusual scaling laws” in the revised manuscript.
- (2) We have revised Fig. 5b into Fig. R1 with fitted exponent index.
- (3) We have revised Fig. 4a-b into Fig. R3.
- (4) We have added Fig. R4 and 5 in the revised SOM.

Figure R4: (Now part of Figure S7 in Supp. Inf. File.) Near-field distributions for the best confined mode in plasmonic, and photonic laser cavity. For each case, the near field distribution is calculated with CdSe thickness varying from 0 to 400 nm.

Reviewer’s comments:

8. *What are the slopes of the curves in Figs. 4a,b,c,d, 5a,5b? Can anything be learned from these slopes?*

Our response:

Indeed, we can extract quantitatively meaning from the slopes of the curves in Figs. 4a, b, c, d, 5a, 5b from the original manuscript through comparisons with new simulations, the above derived expressions for the threshold and the lifetime reduction due to the Purcell effect. The product of this analysis is we can explain the origin of the scaling laws observed as being consistent with accepted theory of laser threshold.

In the following details of our reply, we first analyze the slopes of the curves in Fig. 5a and then Figs. 4a-d, because the scaling law of lifetime versus physical volume (Fig. 5a in the manuscript) plays an important role in the scaling laws of threshold and power consumption versus physical volume (Figs. 4a-d).

With the evidences we presented below, we can conclude that (1) the scaling law of lifetime-physical volume is determined by cavity quality factor and mode volume: we

have performed three dimensional full wave calculations for all the plasmonic and photonic laser devices we measured, and the simulation results match well with experimental data. (2) the scaling law of power consumption-physical volume is mainly determined by population inversion term except for photonic lasers with thickness approaching to diffraction limit where the radiation loss dominates.

(I) Analysis on scaling law of lifetime-physical volume shown in Fig. 5a.

Due to Purcell effect, spontaneous emission rate will be accelerated and can be expressed as $R = R_0 + C_0^{-1}Q/V_m$, where R_0 is the background spontaneous emission rate into other modes, Q is the quality factor, V_m is the mode volume and C_0 is a constant. And the spontaneous emission lifetime τ can be expressed as $\tau \approx C_0 \frac{V_m}{Q}$ providing there is a strong Purcell enhancement.

In order to understand the results of Fig. 5a, we have performed three dimensional full wave calculations for all the plasmonic and photonic laser devices that we measured in order to estimate the Purcell effect. As the TM_{00} mode and TE_{00} mode are with the strongest field confinement and highest effective refractive index in plasmonic and photonic cavities respectively, we focus on the total internal reflection modes of them in our calculations. Two key parameters for an eigenmode of a cavity, quality factor and mode volume are extracted for all the devices as shown in Fig. R5 a and b.

Fig. R5a shows mode volume scaling laws for plasmonic and photonic laser cavities. Clearly, we can see that: (1) the mode volume of a plasmonic cavity is about one order of magnitude smaller than photonic ones at a same physical volume. Such a fact is due to the strong plasmonic field localization effect at the metal-insulator-semiconductor interface; (2) the mode volume of a plasmonic cavity changes slowly with physical volume than photonic ones.

Fig. R5b shows quality factor scaling laws for plasmonic and photonic laser cavities. For photonic cavities, the scaling of Q versus V can be divided into two ranges. At small V , Q is mainly determined by radiation loss and increases with V with an

exponent index of 0.68. At large V , radiation loss becomes smaller, and Q is mainly limited by the self-absorption of CdSe. As a result, the growth rate of Q with V slows down with an exponent index of 0.30. For plasmonic laser cavities, Q is limited by the metallic loss and is inert to the size change. The simulation result gives an exponent index of 0.1.

Figure R5: (Added as Figure S11 in Supp. Inf. File.) Quality factor and mode volume scaling laws. Three dimensional full wave simulations are carried out to make a detail comparison between Q factor and V_{mode} scaling laws as a function of device volume for all the measured plasmonic and photonic lasers cavities. (a-b) Quality factor and mode volume versus CdSe physical volume for plasmonic and photonic lasers cavities. (c-d) Mode volume over Quality factor (V_m/Q) versus CdSe physical volume. The spontaneous emission lifetime, τ should be proportional to V_m/Q according to Purcell effect. Here, all values of Q are for total internal reflection modes of TM_{00} mode and TE_{00} mode, which are with strongest field confinement and highest effective refractive index in plasmonic and photonic cavities respectively. Error bar shows standard deviation.

Fig. R5c and d show scaling laws of $\frac{V_m}{Q}$ for photonic and plasmonic cavities, which are calculated based on Fig. R6a and b. For photonic cavities, $\frac{V_m}{Q}$ changes with physical volume with an exponent index of 0.22 and 0.60 at small and large volume ranges respectively. For plasmonic lasers, the exponent index is 0.53. As mentioned above, the lifetime τ approximately equals to $C_0 \frac{V_m}{Q}$, so the relationship between $\frac{V_m}{Q}$ and physical volume corresponds to the scaling laws of lifetime with volume. We have plotted these simulated scaling laws of $\frac{V_m}{Q}$ with the experimentally measured lifetime scaling law data as shown in Fig. R6. We can see that the simulation results match well with the experimental lifetime data in all volume ranges for photonic lasers and small volume range ($< \sim 5 \lambda^3$) for plasmonic ones. When the volume is larger than $\sim 5 \lambda^3$, the experimental data for plasmonic lasers shifts away from the simulated ones which may be due to higher order hybrid plasmonic modes as mentioned in the original manuscript.

Fig. R6: (New Fig. 5a in the revised manuscript) Scaling laws of lifetime as a function of plasmonic and photonic nanolasers physical volume. Error bar shows standard deviation. Lines are guides to the eye showing simulated relationship between lifetime and physical volume.

(II) Analysis on scaling law of lifetime-threshold shown in Fig. 4.

We analyze the slopes of the curves in Fig. 4 (a-d).

As the threshold power density (shown in Fig. 4a,b in main manuscript) can be directly obtained by dividing power consumption at the threshold (shown in Fig. 4c,d) over the device area, so in the following we focus on the slopes of curves of power consumption at threshold shown in Fig. 4c,d.

The power consumption at threshold can be calculated by $P_{th} \times A$, which results in:

$$P_{th}^{(power)} = \frac{h\nu}{\eta} \left[\frac{1 - \eta_i \beta}{\eta_i \Gamma \beta} \cdot \gamma + R_{non} n_0 V + (1 - \beta) n_0 V \cdot \frac{1}{\tau} \right] \quad (5)$$

Here γ can be substituted by $\frac{\omega}{Q}$, and spontaneous emission lifetime τ can be substituted by $\tau \approx C_0 \frac{V_m}{Q}$. The conversion efficiency of pump photons into electrons η and carrier density at transparency n_0 can be regarded as constants. Again, we only focus on the first and the last term, due to the negligible nonradiative emission. As a result, $P_{th}^{(power)}$ can be expressed as

$$P_{th}^{(power)} \approx C_1 \cdot \frac{1 - \beta}{\Gamma \beta} \cdot \frac{1}{Q} + C_2 \cdot (1 - \beta) \cdot \frac{VQ}{V_m} \quad (6)$$

Here C_1 and C_2 can be regarded as constants which are related with η , ω and n_0 , C_0 respectively. The rest of the parameters Γ, β, Q, V_m are all related to physical volume, which causes the different scaling behaviors of power consumption for plasmonic and photonic lasers.

(1) For both plasmonic and photonic lasers with $T > 150$

First, for both plasmonic and photonic lasers with $T > 150$, the exponent indexes are all positive and similar in values as shown in Fig. 4e and d. The positive exponent indexes indicate that the power consumption at threshold follows the general trend that it reduces with reducing gain volume.

Providing the second term of equation (6) dominates the $P_{th}^{(power)}$, using the scaling of the Q and V_m shown in Fig. R5, we find $P_{th}^{(power)} \sim V^{1+0.1-0.63} = V^{0.47}$ for

plasmonic lasers and $P_{th} \sim V^{1+0.35-0.9} = V^{0.45}$ for photonic ones, which is consistent with the observed values in Fig 4e. We note that here we use only one exponent to fit the quality factor versus physical volume which gives a scaling law of $Q \sim V^{0.35}$. Such an observation that the second term of equation (6) (the term accounts for carrier population inversion) dominates the threshold in this thickness range is consistent with the scaling law of lifetime versus threshold shown in Fig. R1.

(2) *For photonic lasers with $T < 150$*

For the photonic lasers with thickness range from 100 nm to 150 nm and volume smaller than $5\lambda^3$, the power consumption rises rapidly. Here the power consumption scaling with volume has a negative exponent and this must be driven by the increased cavity loss, which is in the first term of the power consumption expression of equation (6), the same mechanism observed previously in the scaling law of lifetime versus threshold shown in Fig. R1.

(3) *For plasmonic lasers with $T < 150$*

When the thickness becomes thinner than 150 nm, the power consumption decreases more dramatically with physical volume, which should be due to the increased β . For thinner devices, the coupling between the photon excited electron-hole pairs and lasing plasmonic mode becomes stronger which will lead to a large β . When considering an increasing β with the decreasing of physical volume in equation (6), both the first and second term will give a smaller power consumption. As a result, the reduction of power consumption will be more rapidly with the decrease of the physical volume.

We have made the following changes in the revised manuscript to clarify this:

- (1) We have added a section of “Rate equation analysis on the unusual scaling laws” in the revised manuscript.
- (2) We have added the simulation results in Fig. R6 c-d into Fig. 5a in the revised manuscript.

(3) We have added Fig. R7 into the revised manuscript as new Fig. 3b.

(4) We have revised Fig. 5a in the original manuscript into Fig. R8.

Response Letter to Reviewer 2

We thank the reviewer 2 for the encouraging remarks of our work as “*This systematic study and comparison of photonic and plasmonic lasers will be highly appreciated by the community.*”, and “*Once these major concerns are addressed, the paper will provide important and critical insight for this increasingly interesting topic.*” By carefully conducting new simulation and data analysis, we are now able to address all of the reviewer’s comments. We believe that these questions have helped us improve the quality and presentation of the manuscript.

Reviewer’s general comments:

In general, the authors should provide more physical interpretations of their results to enhance their experimental data analysis. For example, the intrinsic cavity modes and near-field distributions in both plasmonic and photonic lasers need to be clarified. The authors should provide more theoretical analysis on why 53 is the critical volume number. The mode delocalization with decreased cavity size need to be further verified by near-field distribution results (page 7, Figure S19).

Our responses:

This is an important aspect of this work. We thank the reviewer for asking this question, which helps us to improve the clarity of this part in the manuscript.

In the following, we systematically illustrate the intrinsic cavity modes and near-field distributions in both plasmonic and photonic lasers. These results will clearly reveal the mode delocalization with decreased cavity size. Combining with the scaling law of device thickness versus physical volume, we provide theoretical analysis on why $5\lambda^3$ is the critical volume number (Here we suppose the number of “53” the reviewer mentioned to be a typo of $5\lambda^3$).

Because the reviewer also asked how photonic lasers will perform when other substrates such as sapphire are used. Here we consider three kinds of substrates, including Au/MgF₂, SiO₂ that are used in the experiment, and Al₂O₃ the reviewer

specified. In the following, we will note these three configurations of CdSe/Au/MgF₂, CdSe/SiO₂ and CdSe/Al₂O₃ as plasmonic, photonic I and photonic II configurations.

(1) *near-field distributions*

First, we have calculated *near-field distributions* of these three configurations, where the lateral widths of the waveguide is considered as infinity. As shown in Fig. R1, for each configurations, we calculate the lowest four orders of propagating electromagnetic modes existing in the system with CdSe thickness varying from 0 to 400 nm. Note that, any of these propagating electromagnetic modes will cut off when their effective refractive index approaches to that of its substrate. The refractive indexes of Au/MgF₂, SiO₂ and Al₂O₃, substrates are ~ 1 , ~ 1.47 and ~ 1.76 respectively.

Fig. R1. (Added as Figure S7 in Supp. Inf. File.) Near-field distributions of the four lowest four orders of propagating electromagnetic modes for plasmonic, and photonic I and II configurations. For all the panels, x-axis indicates the position in the unit of nanometer; y-axis indicates the thickness of CdSe in the unit of nanometer. For each mode, we have calculated the near-field distribution for CdSe thickness from 0 to 400 nm. The green lines in each panel are guidance for the eye to track the changing CdSe thickness. TM modes have dominant electrical field perpendicular to the CdSe and substrate interface. TE modes have dominant electrical field parallel to the CdSe and substrate interface.

From these results, we can conclude that: (a) All modes will become delocalized with decreasing of CdSe thickness; (b) For the plasmonic configuration, the TM_{00} has the best field confinement at any given thickness; (c) For the photonic configuration, the TE_{00} of photonic I has the best field confinement at any given thickness; (d) The fundamental plasmonic mode (TM_{00}) has the best field confinement at any given thickness.

(2) *“why $5\lambda^3$ is the critical volume number”*

The reviewer should note the curve in Fig. 4a from the original manuscript with thickness in the range of 100-150 nm. At this thickness range, the effective refractive index of the laser cavity mode is just above cavity total internal reflection cut off thickness even for the most confined TE_{00} mode. The quality factor of the cavity is very sensitive to both thickness and area change as shown in Fig. R2. From Fig. 4a, when the physical volume changes from about $5\lambda^3$ to about $2\lambda^3$ for the thickness in the range of 100-150 nm, the threshold increases about one order of magnitude. Such a volume change corresponds to an area change from 11 (17) μm^2 to 4 (7) μm^2 for the thickness of 150 (100) nm. As shown in Fig. R2, the quality factor of the cavity decreases dramatically in this range. As the total cavity loss rate (γ) is inversely proportional to quality factor (Q), such a fast decrease of Q should be the main reason for the dramatically degrading photonic laser performance. Practically, for the devices with thickness in the range of 100-150 nm, there is already a significant amount of electrical

field localized outside CdSe for the most confined TE₀₀ mode, which will further reduce the quality factor due to surface roughness and any cavity irregularity inducing scattering, which will consequently constrain the scaling down of photonic lasers.

Fig. R2. (Added as Figure S12 in Supp. Inf. File.) Scaling of quality factor Q with physical volume for plasmonic and photonic laser cavity modes. For each case, the quality factor Q are calculated with CdSe nanosquare with thickness of 60 nm, 100 nm, 150 nm and 400 nm respectively. The dotted lines in both figures indicate the same Q value. Here, all values of Q are for total internal reflection modes of TM₀₀ mode and TE₀₀ mode, which are with strongest field confinement and highest effective refractive index in plasmonic and photonic cavities respectively.

We have added Fig. R1 and R2 in the revised supporting materials as well as corresponding discussions in the revised manuscript to clarify this.

Reviewer’s general comments:

Additionally, the lateral dimension and disk thicknesses were not differentiated in their comparison among different physical volumes. For example, the crystal thickness of nano-squares with the same volume ranged from 70 nm to 400 nm (Figure 3a). Especially for thick nano-disks, higher order waveguide mode in the vertical direction can also form within the cavity and provide additional optical feedback for lasing action.

Our responses:

We thank the reviewer for his/her comment. For squares studied here, the lateral dimension is much larger than the thickness, and consequently, they have different weight of impact on the laser performance. For the threshold-physical volume scaling law shown in Fig. 4 of the original manuscript, we have already *differentiated* the lateral dimension and disk thicknesses by categorizing the devices thickness.

For a more detailed analysis on the intrinsic cavity modes, we kindly refer the reviewer to our response to his/her technical comment 3.

In addition, we have added a new panel in the revised manuscript for a clear illustration of square area scaling with volume (Fig. R3).

Figure R3: **(Added as Fig. 3b in the revised manuscript)** The relationship between CdSe area and physical volume for only the *lasing* plasmonic and photonic cavities.

Reviewer’s general comments:

Also, the authors did not provide the rationale underlying their design considerations of photonic lasers. For example, why do they use silica as the substrate? Will the threshold and energy consumption behavior change if other substrates such as sapphire are used, which has higher thermal conductivity and could cool down the device? Also, the refractive index of different substrates will influence the field confinement and device performance of photonic nanolasers.

Our responses:

In the case of photonic devices, the nanosquares need to be placed onto some substrate. While a higher refractive index substrate like sapphire might aid thermal conduction, it would unfortunately increase the critical device thickness of mode cut-off (See Fig. R1). Here, we have minimized thermal heating of the samples by our choice of pumping condition. So this gives us the opportunity to use silica as a substrate, which has high optical quality and low refractive index. Compared to sapphire with a refractive index of ~ 1.76 , silica only has a refractive index of ~ 1.47 , which gives better optical confinement in the CdSe gain material as shown in Fig. R1. The choice of silica substrate ensures the lowest possible cut off thickness/volume.

We have added the following sentence to the revised manuscript to clarify this:

“SiO₂ substrates are choice here for its low refractive index, which gives better optical confinement in gain material.”

Reviewer’s comments:

1. The authors should specify that the record low lasing thresholds are those compared to other hybrid plasmonic waveguide lasers. For plasmonic lasers based on nanoparticle arrays [Nature Nanotechnology 8, 506–511 (2013).], lasing thresholds are around 0.2 W cm⁻² (0.2 mJ cm⁻² with 1 kHz operation), which is lower than the number reported (10 KW cm⁻²) in this manuscript. These two different systems should be differentiated in the background section.

Our responses:

The reviewer comments that the thresholds reported in our work are not as low as those reported in another work. We note that the values specified in our work is the peak pulse intensity, which is necessary to specify the threshold of a pulse pumped laser. The value of 0.2 W cm⁻² from the work that reviewer quoted is the average intensity at threshold and is not an indicator of the laser threshold intensity. However, we can calculate the peak pulse power intensity of the threshold in the work that reviewer

quoted using information from that paper. The laser threshold from the work reviewer quoted is $\sim 0.2 \text{ mJcm}^{-2}$. This value corresponds to the pulse energy density. Using the pulse width quoted in the paper (40 fs), the threshold in terms of peak pulse power intensity is 5 GW cm^{-2} , which is about 5 orders of magnitude higher than the value presented in our work.

However, we agree with the reviewer that, “*these two different systems should be differentiated in the background section*”, as the work that the reviewer quoted is based on a collectively oscillating metallic cavities. To make this clearer to readers, we have revised the corresponding sentence in the background section, which now reads,

“To date, there are numerous reports on laser construction based on a metallic cavity [6-28], and collectively oscillated metallic cavities [29-33].”

Reviewer’s comments:

2. The authors should explain more why the record low lasing threshold was achieved. Is it because of the polycrystalline Au film with smoother surfaces, or improved quantum efficiency of CdSe? Such information will be beneficial for the nanophotonics community

Our response:

We thank the reviewer for his/her comment. The intrinsic reasons for the record low lasing threshold achieved here are three folds: (1) improved quantum efficiency of CdSe that approaches to 100%; (2) improved polycrystalline Au film with smoother surfaces and a high figure of merit ($\frac{-\text{Re}[\epsilon_m]}{\text{Im}[\epsilon_m]}$) of 16; (3) total internal reflection cavity mode with low radiation loss.

To make this clearer to readers, we have added a paragraph in the revised manuscript, which reads,

“The record low threshold achieved here has three intrinsic reasons: (I) improved internal quantum efficiency of CdSe approaches 100%; (II) the polycrystalline Au film

has with smoother surfaces with a high material figure of merit of 16; (III) total internal reflection cavity modes have with low radiation loss.”

Reviewer’s comments:

3. More systematic studies are needed on the near-field distribution of the intrinsic cavity modes. The authors claimed that taller nano-squares can support stronger photonic resonances, but only a few cavity sizes were analyzed, and the near-field distribution of taller disks is missing. In plasmonic nanolasers, larger physical sizes can also lead to higher-order modes, which is possibly the reason for decreased lasing threshold at large physical volumes (Figure 5a).

Our response:

We thank the reviewer for his/her comment. In the following detailed reply, we consider a number of properties of the laser modes in plasmonic and photonic cavities. This analysis has been incorporated in both the manuscript and supplementary information files. Here we consider simulated cavity mode characteristics such as effective index of transverse modes, cavity mode volume and Q-factors. We also discuss how these parameters scale with physical volume, which allows us to compare this new simulation study with experimental results in the manuscript.

We have systematically studied the near-field distribution of the intrinsic cavity modes for both plasmonic and photonic cavities as shown in Fig. R1 as above. The thickness range we studied is from 0 nm to 400 nm, covering most of the laser devices. We show near-field distributions for plasmonic, and photonic cavity modes for all the lowest four orders of propagating electromagnetic modes of TM_{00} , TE_{00} , TM_{01} and TE_{01} . These results indicate that while all modes will become delocalized with decreasing of CdSe thickness, the fundamental plasmonic mode (TM_{00}) has the best field confinement at any given thickness.

We have also calculated the *effective refractive indexes* of all three configurations, which are shown in Fig. R4. The fundamental plasmonic mode (TM_{00}) has the largest

effective refractive index at any given thickness among all the modes, which gives the best field confinement for resonant cavity feedback. For the photonic configurations, fundamental, the fundamental TE₀₀ mode has the best field confinement.

In both plasmonic and photonic nanosquare cavities, the most efficient feedback is based on total internal reflection (TIR), which has a cavity mirror loss coefficient about two orders of magnitude lower than that of Fabry–Pérot feedback [refs. 11, 16, 44, 45 in the revised manuscript]. The lowest order of TIR feedback in a nanosquare cavity requires a critical angle of TIR, $\theta_c \geq 45^\circ$, which requires the effective refractive index of the cavity mode, $n_{cavity} \geq \sqrt{2} n_{surrendering}$. Here, $n_{surrendering}$ can be approximately treated as substrate refractive index.

Fig. R4. (Added as Figure S6 in Supp. Inf. File.) Effective refractive indexes for the lowest four orders of modes in CdSe-MgF₂/Au, CdSe-SiO₂ and CdSe-Al₂O₃ configurations. Dashed lines in panels indicate the lowest effective refractive index to support a total internal reflection cavity mode.

From Fig. R1 and R3, we can see that the fundamental plasmonic TM₀₀ mode has the best field confinement and the largest effective refractive index due to its strong hybridization with surface plasmon mode. And in the photonic configurations, the fundamental TE₀₀ mode has the best field confinement and the largest effective refractive index. Both the strong field confinement and high effective refractive index are crucial for lasing performance.

We have further performed three dimensional full wave calculations for all the plasmonic and photonic laser devices we measured. As the TM_{00} mode and TE_{00} mode have with strongest field confinement and highest effective refractive index in plasmonic and photonic cavities respectively, we focus on the total internal reflection mode of them in the three dimensional calculations. Two key parameters for an eigenmode of a cavity, quality factor and mode volume are extracted for all the devices as shown in Fig. R5.

Figure R5: (Added as Figure S11 in Supp. Inf. File.) Quality factor and mode volume scaling laws. Three dimensional full wave simulations are carried out to make a detail comparison between Q factor and V_{mode} scaling laws as a function of device volume for all the measured plasmonic and photonic lasers cavities. (a-b) Quality factor and mode volume versus CdSe physical volume for plasmonic and photonic lasers cavities. (c-d) Mode volume over Quality factor (V_m/Q) versus CdSe physical volume. The spontaneous emission lifetime, τ should be proportional to V_m/Q according to Purcell effect. Here, all values of Q are for total internal reflection modes of TM_{00} mode and TE_{00} mode, which are with strongest field confinement and highest effective refractive index in plasmonic and photonic cavities respectively. Error bar shows standard deviation.

Fig. R5a shows mode volume scaling laws for plasmonic and photonic laser cavities. Clearly, we can see that: (1) the mode volume of a plasmonic cavity is about one order of magnitude smaller than photonic ones at a same physical volume. Such a fact is due to the strong plasmonic field localization effect at the metal-insulator-semiconductor interface; (2) the mode volume of a plasmonic cavity changes slowly with physical volume than photonic ones.

Fig. R5b shows quality factor scaling laws for plasmonic and photonic laser cavities. For photonic cavities, the scaling of Q versus V can be divided into two ranges. At small V , Q is mainly determined by radiation loss and increases with V with an exponent index of 0.68. At large V , radiation loss becomes smaller, and Q is mainly limited by the self-absorption of CdSe. As a result, the growth rate of Q with V slows down with an exponent index of 0.30. For plasmonic laser cavities, Q is limited by the metallic loss and is inert to the size change. The simulation result gives an exponent index of 0.1.

Fig. R5c and d show scaling laws of $\frac{V_m}{Q}$ for photonic and plasmonic cavities, which are calculated based on Fig. R6a and b. For photonic cavities, $\frac{V_m}{Q}$ changes with physical volume with an exponent index of 0.22 and 0.60 at small and large volume ranges respectively. For plasmonic lasers, the exponent index is 0.53. As mentioned above, the lifetime τ approximately equals to $C_0 \frac{V_m}{Q}$, so the relationship between $\frac{V_m}{Q}$ and physical volume corresponds to the scaling laws of lifetime with volume. We have plotted these simulated scaling laws of $\frac{V_m}{Q}$ with the experimentally measured lifetime scaling law data as shown in Fig. R6. We can see that the simulation results match well with the experimental lifetime data in all volume ranges for photonic lasers and small volume range ($< \sim 5 \lambda^3$) for plasmonic ones. When the volume is larger than $\sim 5 \lambda^3$, the experimental data for plasmonic lasers shifts away from the simulated ones which may be due to higher order hybrid plasmonic modes as mentioned in the original manuscript.

Fig. R6: (New Fig. 5a in the revised manuscript) Scaling laws of lifetime as a function of plasmonic and photonic nanolasers physical volume. Error bar shows standard deviation. Lines are guides to the eye showing simulated relationship between lifetime and physical volume.

We have made the following changes in the revised manuscript to clarify this:

- (5) We have added the simulation results shown in the left two panels of Fig. R4 in the revised supporting information.
- (6) We have added the simulation results in Fig. R5 c-d into Fig. 5a in the revised manuscript.
- (7) We have revised Fig. 5a in the original manuscript into Fig. R6.

Reviewer's comments:

4. The authors should justify their mode volume calculations considering the intrinsic divergent nature of plasmonic modes. Related theoretical analysis can be found in *ACS Photonics* 1, 2-10 (2014) and *Physical Review Letters* 110, 237401 (2013).

Our response:

We thank the reviewer for his/her comment. The method used in the manuscript to calculate mode volume is the customary definition of effective mode volume, which is

the magnitude of the energy-density volume integration over the maximal energy density. Such a definition is suitable for Hermitian system without loss, where the solutions of Maxwell equations are normal modes, which approach zero for $r \rightarrow \infty$. For dissipative cavities, the solutions are quasinormal modes (QNMs) as the reviewer remarks and this requires a different treatment for accuracy. The outgoing wave boundary conditions force the QNMs at $r \rightarrow \infty$ behave as outgoing waves of the form $\tilde{f}(\vec{r}) \propto \frac{\exp(ik_0 r)}{r}$, where $r = |\vec{r}|$. Since $k_0 = k_0^R + ik_0^I$ with $k_0^I < 0$, they diverge exponentially as $r \rightarrow \infty$. (*ACS Photonics* 1, 2-10 (2014); *Physical Review Letters* 110, 237401 (2013)). Consequently, when performing infinite volume integration, the energy tends to diverge when using the customary definition, described above.

However, the simple customary method can still be applied by integrating over a finite volume. Provided the field decays away from the cavity sufficiently quickly this can be an accurate evaluation and this usually depends in three dimensional full wave simulations by the computing power and memory available. In this case, the results from the two calculation methods tend to be similar. For instance, the mode volumes of a photonic crystal cavity calculated by the two methods only differs by about 5% for a calculation field as large as six times of lattice constant (*ACS Photonics* 1, 2-10 (2014)).

When the divergent is considered, the mode volume may be overestimated in the customary definition, especially for plasmonic cavities. Although this will not affect any claim for the current work, we have added the following sentence in the revised supporting information to clarify this.

“We note that the mode volume calculated here may be overestimated due to the inherent field divergence of leaky cavity eigenmodes [1-2].”

Reviewer’s comments:

5. Can the authors perform more theoretical analysis to predict the lasing threshold for plasmonic and photonic cases? Their current theoretical analysis focused on quality

factors and mode volumes of the cold cavity. More gain-loss analysis by incorporating gain within the modeling is essential to verify the trends observed in experiments and to reveal the mechanism behind the unusual scaling laws.

Our response:

This is an important aspect of this work. Although the quality factors and mode volumes are main parameters that determines laser threshold, there are indeed other parameters that should be considered for a more rigorous analysis. Here, we conducted rate equation analysis to build the relationship between threshold and other key laser parameters: We thank the reviewer for asking this question, which helps us to improve the clarity of this part in the manuscript.

We first start with semiconductor rate equations that follow:

$$\frac{dN_2}{dt} = \eta p - R_{\text{non}}N_2 - RN_2 - \Gamma R \beta(N_2 - N_0)N_{\text{ph}} \quad (1)$$

$$\frac{dN_{\text{ph}}}{dt} = -\gamma N_{\text{ph}} + \beta RN_2 + \Gamma R \beta(N_2 - N_0)N_{\text{ph}} \quad (2)$$

Here, N_2 is the excited carrier population, N_{ph} is the photon number of a single laser, p is the pump rate, N_0 is the excited state population at transparency, η is the conversion efficiency of pump photons into electron/hole pairs, R_{non} is the nonradiative recombination rate, R is the spontaneous emission rate, Γ is the confinement factor, β is the spontaneous emission coupling factor, γ is the total cavity loss rate.

The first equation describes the rate of change of the carrier population, while the second equation describes the rate of change of the photon number. In the quasi steady state conditions considered in our experiments, $\frac{dN_2}{dt} = 0$ and $\frac{dN_{\text{ph}}}{dt} = 0$, the photon number N_{ph} can be described by the quadratic equation:

$$\gamma N_{\text{ph}}^2 - \left[\eta p - \frac{R_{\text{non}} + R}{\Gamma R \beta} \gamma - (R_1 + R)N_0 + \beta RN_0 \right] N_{\text{ph}} - \frac{\eta p}{\Gamma} = 0 \quad (3)$$

The solution of the above equation has two linear trends at large and small pump rates

respectively, which define the conventional kink in the light-light curve. We define the threshold at the value of power density where these linear trends intersect. The pump power density is related to pump rate by $P = R_p h\nu/A$, where $h\nu$ is the photon energy and A is the device area. The threshold P_{th} is thus,

$$P_{th} = \frac{h\nu}{\eta} \left[\frac{1 - \eta_i \beta}{\eta_i \Gamma \beta A} \cdot \gamma + R_{non} n_0 T + (1 - \beta) n_0 T \cdot \frac{1}{\tau} \right] \quad (4)$$

Here, $\eta_i = \frac{R}{R_{non} + R} \approx 1$ is the internal quantum efficiency of the gain material, so that $R_{non} \ll R$. $\tau \approx \frac{1}{R}$ is the spontaneous emission lifetime. $n_0 = \frac{N_0}{V}$ is the excited state density at transparency where V is the volume of the CdSe. $T=V/A$ is the thickness of the CdSe nanosquare.

The first term of P_{th} is proportional to γ inversely proportional to Q accounting for the cavity photon loss compensation. The second term is proportional to R_{non} , accounting for nonradiative carrier loss. The last term, which is proportional to n_0/τ , accounting for the power density required to achieve carrier population inversion.

The debated role of Purcell effect on the laser threshold is clear in equation (3). Firstly, the Purcell effect increases β , which reduces the first and last terms. However, accelerated spontaneous emission reduces τ , making the last term larger. Because the internal quantum efficiency is close to unity for both plasmonic and photonic lasers (nonradiative emission is negligible), in the following, we focus on the first and the last terms here.

We can see that, for a giving thickness T , P_{th} should be inversely proportional to τ providing the last term dominants the P_{th} and β is significantly small then unit.

Below, we use equations (4) to analyze the experimental scaling laws presented in Fig. 4 and 5 in the original manuscript, where we can see that the mechanism behind the unusual scaling laws can be well illustrated.

We will start with the analysis on Fig. 5b, and then Fig. 5a, and lastly Figs. 4a-d in the original manuscript.

(I) Analysis on scaling law of lifetime-threshold shown in Fig. 5b.

To analyze the mechanism behind the unusual scaling law of lifetime-threshold, and to clarify the origin of that plasmonic lasers can have simultaneously shorter lifetime and lower threshold, we further analyzed the relationship between lifetime and threshold by categorizing the data in Fig. 5b of the original manuscript by nanosquare thickness. As shown in Fig. R7, the difference between plasmonic and photonic lasers becomes more distinct for thinner devices.

Figure R7: (Added as Fig. 5b-d in the revised manuscript) Scaling of lifetime with threshold for plasmonic (black dots) and photonic (red dots) lasers categorized by CdSe nanosquare thickness as marked in the panels. Solid lines are fitted lines.

We now compare the theoretical predicted thresholds from equation (4) with experimental data. For each thickness range, we empirically fit the experiment data in Fig. R1 by $\tau \propto P_{th}^{\alpha_1}$, where α_1 is an exponent and is depicted in Fig. R1 .

We can see that, (I) for all of thickness ranges of plasmonic lasers and thickness thicker than 150 nm of photonic lasers, α_1 is close to but larger than -1, which indicates that the last term of equation (3) dominates over the first term. We note that, for both plasmonic and photonic lasers, the absorption from the gain material should contribute a significant portion to the laser threshold. And clearly, the shortened τ induced by small cavity dominates the scaling law in this range. (II) α_1 for photonic lasers become significantly larger than that for plasmonic lasers when the thickness of CdSe is between 100-150 nm.

In order to gain further insight into this phenomenon, we must also analyze the

relationship between threshold and physical volume, as shown in Fig. 4 a and b in the original manuscript (and Fig. R8 replotted here in a linear scale). Importantly, the threshold of photonic lasers will dramatically increase with the decrease of physical volume when the thickness of CdSe is between 100-150 nm.

Figure R8: (New Fig. 4a and b in the revised manuscript) Scaling of threshold with physical volume in linear scale for plasmonic (black dots) and photonic (red dots) lasers categorized by CdSe nanosquare thickness as marked in the panels.

From Fig. R7 and R8, we can see a general trend for both plasmonic and photonic lasers that a smaller physical volume and a shorter lifetime will result in a higher threshold. But for the lasers with thicknesses approaching the diffraction limit, plasmonic lasers clearly have simultaneously shorter lifetime and lower threshold. Among the parameters in the first and last terms of equation (1), the total cavity loss rate, γ should be the fact that causes such a dramatic performance degradation of photonic lasers compared to plasmonic ones. The cavity loss γ can be expressed as $\frac{\omega}{Q}$, where Q is the quality factor and ω is the resonant frequency of the cavity. For a giving thickness range, all other parameters should not change dramatically with the reduction of the side length and consequently physical volume to a few λ^3 . However, Q of a photonic cavity is very sensitive to the thickness or side length change in this range as shown in Fig. R2.

Practically, the decreased confinement factor Γ in the thickness range approaching the diffraction limit should be another factor for the poor performance of photonic lasers.

As shown in Fig. R1, for thinner devices, photonic cavity has a more delocalized mode compared to the plasmonic cavity, which results in a quality factor that is more sensitive to the surface roughness and cavity irregularities.

For plasmonic lasers, the field is well confined (Fig. R1) and their quality factor is inert to the size change till the thickness approaches tens of nanometer (Fig. R2). And thus they can have higher Purcell factor (shorter lifetime) and lower threshold simultaneously comparing to photonic lasers at small size.

(II) Analysis on scaling law of lifetime-physical volume shown in Fig. 5a.

The detailed analysis on scaling law of lifetime-physical volume shown in Fig. 5 has been provided in our response to reviewer's technical comment 3. We kindly refer the reviewer to see our response to his/her technical comment 3.

(III) Analysis on scaling law of lifetime-threshold shown in Fig. 4.

As the threshold power density (shown in Fig. 4a,b in main manuscript) can be directly obtained by dividing power consumption at the threshold (shown in Fig. 4c,d) over the device area, so in the following we focus on the slopes of curves of power consumption at threshold shown in Fig. 4c,d.

The power consumption at threshold can be calculated by $P_{th} \times A$, which results in:

$$P_{th}^{(power)} = \frac{h\nu}{\eta} \left[\frac{1 - \eta_i \beta}{\eta_i \Gamma \beta} \cdot \gamma + R_{non} n_0 V + (1 - \beta) n_0 V \cdot \frac{1}{\tau} \right] \quad (5)$$

Here γ can be substituted by $\frac{\omega}{Q}$, and spontaneous emission lifetime τ can be substituted by $\tau \approx C_0 \frac{V_m}{Q}$. The conversion efficiency of pump photons into electrons η and carrier density at transparency n_0 can be regarded as constants. Again, we only focus on the first and the last term, due to the negligible nonradiative emission. As a result, $P_{th}^{(power)}$ can be expressed as

$$P_{th}^{(power)} \approx C_1 \cdot \frac{1 - \beta}{\Gamma \beta} \cdot \frac{1}{Q} + C_2 \cdot (1 - \beta) \cdot \frac{VQ}{V_m} \quad (6)$$

Here C_1 and C_2 can be regarded as constants which are related with η , ω and n_0 , C_0 respectively. The rest of the parameters Γ , β , Q , V_m are all related to physical volume, which causes the different scaling behaviors of power consumption for plasmonic and photonic lasers.

(4) For both plasmonic and photonic lasers with $T > 150$

First, for both plasmonic and photonic lasers with $T > 150$, the exponent indexes are all positive and similar in values as shown in Fig. 4e and d. The positive exponent indexes indicate that the power consumption at threshold follows the general trend that it reduces with reducing gain volume.

Providing the second term of equation (6) dominates the $P_{th}^{(power)}$, using the scaling of the Q and V_m shown in Fig. R5, we find $P_{th}^{(power)} \sim V^{1+0.1-0.63} = V^{0.47}$ for plasmonic lasers and $P_{th} \sim V^{1+0.35-0.9} = V^{0.45}$ for photonic ones, which is consistent with the observed values in Fig 4e. We note that here we use only one exponent to fit the quality factor versus physical volume which gives a scaling law of $Q \sim V^{0.35}$. Such an observation that the second term of equation (6) (the term accounts for carrier population inversion) dominates the threshold in this thickness range is consistent with the scaling law of lifetime versus threshold shown in Fig. R1.

(5) For photonic lasers with $T < 150$

For the photonic lasers with thickness range from 100 nm to 150 nm and volume smaller than $5\lambda^3$, the power consumption rises rapidly. Here the power consumption scaling with volume has a negative exponent and this must be driven by the increased cavity loss, which is in the first term of the power consumption expression of equation (6), the same mechanism observed previously in the scaling law of lifetime versus threshold shown in Fig. R1.

(6) For plasmonic lasers with $T < 150$

When the thickness becomes thinner than 150 nm, the power consumption decreases more dramatically with physical volume, which should be due to the increased β . For thinner devices, the coupling between the photon excited electron-hole pairs and lasing plasmonic mode becomes stronger which will lead to a large β . When considering an increasing β with the decreasing of physical volume in equation (6), both the first and second term will give a smaller power consumption. As a result, the reduction of power consumption will be more rapidly with the decrease of the physical volume.

We have made the following changes in the revised manuscript to clarify this:

- (5) We have added a section of “Rate equation analysis on the unusual scaling laws” in the revised manuscript.
- (6) We have revised Fig. 5b into Fig. R7 with fitted exponent index.
- (7) We have revised Fig. 4a-b into Fig. R8.

Reviewer’s comments:

6. In general, the figures can be improved for better visualization. For example, the pink curves and text in Figure 4 were hard to resolve, and the figure resolutions (especially those in supplementary information) should be higher.

Our response:

We thank the reviewer for his/her comment. We have now modified the figures in color and resolution for a better visualization. We note that the low resolution of figures presented in the submission system may be due to the online pdf conversion.

Reviewer’s comments:

7. The fitting curves deviated much from experimental data in Figures 4a-d. Can the authors explain the origin of big fluctuations in their measured data?

Our response:

In Fig. 4a-d, data are categorized in thickness ranges but not a fixed thickness, which we think is the main reason for the fluctuations. Other facts, such as differences in internal quantum efficiency, irregular cavity shape will also introduce fluctuations. However, despite with the big fluctuations, the general trend the measured data revealed are clear.

We thank reviewers for their detailed comments on our manuscript. We hope the revisions are satisfactory and the revised manuscript is suited for publication.

Reviewers' comments:

Reviewer #1 (Remarks to the Author):

Review of the revised manuscript "Unusual Scaling Laws for Plasmonic Lasers beyond Diffraction Limit" by S. Wang et al.

The Referee is (almost) satisfied with the authors' explanations and the revision of the manuscript. In particular, the Referee likes the response to the comment related to the beta factor.

At the same time, the Referee recommends to remove the newly added text referencing the beta factor:

"We note that scaling laws for plasmonic lasers with even smaller cavity sizes would require further study. At extremely small cavity size region, where the β factor tends to 1, light-light curve will straighten to be a straight line known as a 'thresholdless' behavior, and a more detailed theory is needed to define a laser's threshold, for example using phase space methods or photon statistics. [46, 47]. At such a situation, the power consumption should be determined dominantly by cavity loss rate."

When this is done, the Referee can recommend this paper to be published.

Reviewer #2 (Remarks to the Author):

The authors have compiled a thorough response in addressing the issues raised by the reviewers, including (1) the rate equation analysis to support the unusual scaling laws on threshold and lasing lifetime and (2) the near-field distributions to indicated mode delocalization for decreased cavity size for both plasmonic and photonic systems. The quantification of Q of different device sizes supported why $5\lambda^3$ is the critical volume number.

Based on the rate equations, the authors proposed scaling laws of lasing threshold and lifetime using the Purcell effect. However, lasing is a nonlinear optical process and does not depend linearly on the mode quality of the passive structure. Also, in this paper, only the spontaneous emission rate and β factor are represented in the rate equation and terms representing stimulated emission and time-dependent lasing buildup are missing. The paper would benefit by clearly stating the limitations of the model. Also, the authors should specify that the lasing threshold is described in terms of peak power instead of average power in the main text. Such information is important for the cross-comparison of different plasmonic lasing systems.

After these minor revisions, we recommend this paper for publication in Nature Communications.

Response Letter to Reviewer 1

Reviewer's comment:

The Referee is (almost) satisfied with the authors' explanations and the revision of the manuscript. In particular, the Referee likes the response to the comment related to the beta factor.

At the same time, the Referee recommends to remove the newly added text referencing the beta factor:

“We note that scaling laws for plasmonic lasers with even smaller cavity sizes would require further study. At extremely small cavity size region, where the β factor tends to 1, light-light curve will straighten to be a straight line known as a ‘thresholdless’ behavior, and a more detailed theory is needed to define a laser’s threshold, for example using phase space methods or photon statistics. [46, 47]. At such a situation, the power consumption should be determined dominantly by cavity loss rate.”

When this is done, the Referee can recommend this paper to be published.

Our responses:

We thank the reviewer for his/her comment. We have removed the newly added text referencing the beta factor in the revised manuscript.

Response Letter to Reviewer 2

Reviewer's comment:

The authors have compiled a thorough response in addressing the issues raised by the reviewers, including (1) the rate equation analysis to support the unusual scaling laws on threshold and lasing lifetime and (2) the near-field distributions to indicated mode delocalization for decreased cavity size for both plasmonic and photonic systems. The quantification of Q of different device sizes supported why $5\lambda^3$ is the critical volume number.

Based on the rate equations, the authors proposed scaling laws of lasing threshold and lifetime using the Purcell effect. However, lasing is a nonlinear optical process and does not depend linearly on the mode quality of the passive structure. Also, in this paper, only the spontaneous emission rate and β factor are represented in the rate equation and terms representing stimulated emission and time-dependent lasing buildup are missing. The paper would benefit by clearly stating the limitations of the model. Also, the authors should specify that the lasing threshold is described in terms of peak power instead of average power in the main text. Such information is important for the cross-comparison of different plasmonic lasing systems.

After these minor revisions, we recommend this paper for publication in Nature Communications.

Our responses:

We thank the reviewer for his/her comment. The threshold analysis in the manuscript are based on the quasi-steady state conditions of the experiments, which does not include the time-dependent lasing buildup process. We have added the following sentence to the revised manuscript to clarify this:

“We note that the above analysis are based on the quasi-steady state conditions of the experiments. A time-dependent lasing buildup process comparison between plasmonic and photonic lasers is also interesting but not yet included.”

We have also added a few sentence to specify that the lasing threshold is described in terms of peak power instead of average power in the main text in the revised manuscript:

“We note that the devices are optically pumped by a nanosecond pump laser with pulse width comparable to the spontaneous emission lifetime of the gain material, which helps to accumulate the excited carriers to achieve population inversion for lasing before they recombine and radiate (see Method). The threshold values specified in this work is the peak pulse intensity of the pump laser, which is necessary to specify the threshold of a pulse pumped laser.”

We thank reviewers for their detailed comments on our manuscript. We hope the revisions are satisfactory and the revised manuscript is suited for publication.